# Gain-of-function mutant p53 together with ERG proto-oncogene drive prostate cancer by beta-catenin activation and pyrimidine synthesis

Donglin Ding[1], Alexandra M. Blee[1,6], Jianong Zhang[1], Yunqian Pan[1], Nicole A. Becker[1], L. James Maher 3rd[1], Rafael Jimenez [2], Liguo Wang [3] ✉ & Haojie Huang [1,4,5] ✉

Whether *TMPRSS2-ERG* fusion and *TP53* gene alteration coordinately promote prostate cancer (PCa) remains unclear. Here we demonstrate that *TMPRSS2-ERG* fusion and *TP53* mutation / deletion co-occur in PCa patient specimens and this co-occurrence accelerates prostatic oncogenesis. p53 gain-of-function (GOF) mutants are now shown to bind to a unique DNA sequence in the *CTNNB1* gene promoter and transactivate its expression. ERG and β-Catenin co-occupy sites at pyrimidine synthesis gene (PSG) loci and promote PSG expression, pyrimidine synthesis and PCa growth. β-Catenin inhibition by small molecule inhibitors or oligonucleotide-based PROTAC suppresses TMPRSS2-ERG- and p53 mutant-positive PCa cell growth in vitro and in mice. Our study identifies a gene transactivation function of GOF mutant p53 and reveals β-Catenin as a transcriptional target gene of p53 GOF mutants and a driver and therapeutic target of TMPRSS2-ERG- and p53 GOF mutant-positive PCa.

The *TP53* tumor suppressor gene is the most frequently mutated gene in human cancers[1–3]. The vast majority of p53 mutants are single amino-acid substitution mutations that are often clustered at certain hotspot resides in the p53 DNA binding domain (DBD) including R175, G245, R248, G249, R273 and R282[3–5]. While losing the canonical tumor suppressor functions in regulation of cell cycle arrest and apoptosis[6–8], the DBD missense mutations are reported to exert dominant-negative effects and acquire additional neomorphic functions and therefore are termed as gain-of-function (GOF) mutations[9–13]. Previous studies have suggested that p53 GOF mutants e activate oncogenic programs via interaction with other transcription factors or co-regulatory proteins such as ETS2 and CREB1[2,10,14–16]. It has also been implied that p53 GOF

mutants can bind certain unusual DNA structures[17,18]. However, whether p53 mutants can directly bind double-stranded DNA in chromatin to contribute to altered gene transactivation has not been explored.

The E26 transformation-specific (ETS)-related gene (ERG) and other ETS family transcription factors are crucial for establishment of different cell lineages (e.g., endothelial vs. hematopoietic) during development[19,20]. These transcription factors induce cell type-specific gene expression by acting as master transcriptional activators and/or form protein complexes with other transcription factors[21,22]. Over-expression of N-terminal truncated, but fully functional ERG due to *ERG* gene fusion with *TMPRSS2* gene occurs in approximately 50% of prostate cancer (PCa) patients[23,24]. Akin to its role in regulating cell

[1]Department of Biochemistry and Molecular Biology, Mayo Clinic College of Medicine and Science, Rochester, MN 55905, USA. [2]Department of Laboratory Medicine and Pathology, Mayo Clinic College of Medicine and Science, Rochester, MN 55905, USA. [3]Division of Biomedical Statistics and Informatics, Mayo Clinic College of Medicine and Science, Rochester, MN 55905, USA. [4]Department of Urology, Mayo Clinic College of Medicine and Science, Rochester, MN 55905, USA. [5]Mayo Clinic Cancer Center, Mayo Clinic College of Medicine and Science, Rochester, MN 55905, USA. [6]Present address: Department of Biochemistry, Vanderbilt University, Nashville, TN 73240, USA. ✉e-mail: wang.liguo@mayo.edu; huang.haojie@mayo.edu

lineage commitment during development, overexpressed ERG has been implicated in aberrant activation of oncogenic programs in PCa through its interactions with master transcription factors such as androgen receptor (AR), HOXB13 and FOXA1[25,26]. While ERG overexpression has long been recognized as a pro-oncogenic factor in PCa[23,24,27], transgenic expression of ERG alone is insufficient to induce prostate tumorigenesis in mice at age up to one year. However, TMPRSS2-ERG transgenic mice do develop aggressive prostate tumors at advanced age (>2 years), highlighting a context-dependent action of TMPRSS2-ERG in PCa pathogenesis[25,28–31]. While both *ERG* and *TP53* genes are frequently altered in PCa patient specimens, the potential for their functional interplay in prostate oncogenesis has remained unknown.

In the present study we reveal that GOF mutant p53 proteins directly bind a unique DNA motif in the *CTNNB1* gene promoter and transactivate *CTNNB1* gene expression. We further show that over-expressed ERG and β-Catenin co-occupy sites in pyrimidine synthesis gene (PSG) loci and promote pyrimidine synthesis and prostate oncogenesis. We also demonstrate that β-Catenin is a viable therapeutic target of TMPRSS2-ERG- and GOF mutant p53-positive PCa.

## Results

### Occurrence of *TMPRSS2-ERG* fusion and *TP53* inactivation in PCa patient specimens

*TMPRSS2-ERG* gene fusion frequently occurs in PCa patients[32,33]. The tumor suppressor gene *TP53* is also often inactivated (loss of function) due to gene deletion and/or mutation in PCa[34,35]. We therefore sought to determine whether these two lesions co-occur in PCa patient samples. Meta-analysis of TCGA data showed that *TMPRSS2-ERG* fusion co-occurred with *TP53* inactivation (including heterozygous and homozygous deletions, gene mutations and other alterations) in primary/localized PCa patient samples[23] (Fig. 1a, b and Supplementary 1a), suggesting that both lesions might be involved in prostate tumorigenesis. Similar results were observed in the SU2C cohort of metastatic PCa[34] (Fig. 1a, b and Supplementary 1b) and the MSKCC cohort including both primary and metastatic PCa[36] (Supplementary Fig. 1c, d). Thus, meta-analysis of data from >1500 PCa patient specimens shows that *TMPRSS2-ERG* fusion and *TP53* inactivation co-occur in PCa patient samples, raising the potential for a cooperative role of these two lesions in prostate tumorigenesis and progression.

### Cooperativity of *TMPRSS2-ERG* transgene with *Trp53* deletion and a GOF role of mutant p53 in prostate oncogenesis

To test the disease relevance of co-occurring *TMPRSS2-ERG* fusion and *TP53* inactivation in PCa patients, we recapitulated this combination by using *Pb*-driven *Cre* recombinase transgenic mice (*Pb-Cre4*)[37], *Pb-TMPRSS2-ERG* (T2-ERG) transgenic mice[38] and *Trp53*[loxp-stop-loxp-R172H/loxp] mice[12] as founding breeders to generate six groups of genetically engineered mice (GEM): 1) wild-type (WT) littermate controls; 2) prostatic cell (PC)-specific *T2-ERG* transgenic (*Pb-T2-ERG*); 3) prostatic cell-specific *Trp53* knockout (*Pb-Cre+;Trp53*[p/p] or *Trp53*[pc/-]); 4) prostatic cell-specific *Trp53* knockout and R172H mutant knockin (*Pb-Cre+;Trp53*[R172H/p] or *Trp53*[pcR172H/-]); 5) prostatic cell-specific *T2-ERG* transgenic and *Trp53* knockout (*Pb-Cre+;Pb-T2-ERG;Trp53*[p/p] or *Pb-T2-ERG;Trp53*[pc/-]); 6) prostatic cell-specific *T2-ERG* transgenic and *Trp53* knockout/knockin (*Pb-Cre+;Pb-T2-ERG;Trp53*[R172H/p] or *Pb-T2-ERG;Trp53*[pcR172H/-]) (Fig. 1c). We employed R172H, a missense mutation in the murine p53 DBD as a surrogate because its equivalent mutation R175H in human p53 is one of the mutations that often occurs in both primary and advanced PCa patient samples (Supplementary Fig. 1e). TP53 R175H is traditionally considered to be a GOF mutation.

As predicted by the corresponding patient data (Fig. 1a, b and Supplementary Fig. 1a–d), *TMPRSS2-ERG* overexpression together with *Trp53* deletion induced focal low-grade prostatic intraepithelial neoplasia (PIN), a precursor of cancerous lesions in the prostate of *Pb-T2-ERG;Trp53*[pc/-] mice as early as 10 months of age (Supplementary

Fig. 2a, b) and high-grade PIN (HGPIN) and focal adenocarcinoma in approximately 50% of 15 month-old mice (Fig. 1c, d). Importantly, *Pb-T2-ERG;Trp53*[pcR172H/-] mice at 10 months of age developed HGPIN and adenocarcinoma (Supplementary Fig. 2a, b) and approximately 90% developed aggressive HGPIN and/or adenocarcinomas by 15 months of age (Fig. 1c, d). Notably, immunohistochemistry (IHC) analysis showed that all the cancerous and PIN lesions in these various strains were AR-positive (Fig. 1c and Supplementary Fig. 2a). Consistent with the previous reports[25,28–31], no PIN was observed in *Pb-T2-ERG* mice by 10 months; However, approximately 20% of these animals displayed focal LGPIN lesions by 15 months of age (Fig. 1d). This observed age-dependent disease progression further supports the notion that ERG overexpression can promote prostate oncogenesis by cooperating with other genetic alterations. The histological changes corresponded with increased Ki67 staining in *Pb-T2-ERG;Trp53*[pc/-] and *Pb-T2-ERG;Trp53*[pcR172H/-] mice at both ages (Fig. 1c, e and Supplementary Fig. 2a, c). Cytokeratin 8 (CK8) and smooth muscle actin (SMA) IHC analyses showed that prostate tumors in *Pb-T2-ERG;Trp53*[pcR172H/-] mice at 15 months were CK8-positive, but SMA staining was not observed (Supplementary Fig. 2d), suggesting that these tumors are luminal type with an invasive phenotype. Together, these data indicate that *TMPRSS2-ERG* overexpression in combination with *TP53* inactivation is sufficient to drive prostate tumorigenesis and that DBD mutant p53 (e.g., GOF mutant R172H) can significantly accelerate PCa oncogenesis, confirming a GOF role for the p53 R172H DBD mutation in promoting disease progression and aggressiveness. This notion is further supported by the observation in patient samples that the incidence of *TP53* gene mutations, including missense, truncating and splice site mutations, was about three-fold higher in advanced PCa in SU2C patients (-36.7%) compared to primary PCa in TCGA patients (-12.5%) (Supplementary Fig. 1c) and by the overall survival data in mice (Fig. 1f). The *Trp53* knockin mouse data are also consistent with the observation in patients that, similar to the total *TP53* mutations, *TP53* DBD mutations also co-occurred with *ERG* fusions in advanced PCa in the SU2C cohort (Supplementary Fig. 1b).

In addition to the R175H mutation, a diverse collection of other hotspot mutations in the DBD of p53 (e.g., R248W and R273H) have been reported in PCa patients[3]. Indeed, the majority of p53 mutations detected in advanced PCa cases[39] are located in the p53 DBD (Supplementary Fig. 1e). In human PCa cell line VCaP (*TMPRSS2-ERG* (ERGΔN39)/*TP53*[R248W/-] DBD mutant-positive) we demonstrated that knockdown of either ERG or DBD mutant p53 alone or together inhibited VCaP cell growth (Fig. 1g, h). Together, these data indicate that mutations such as R172H in murine *Trp53* and R248W in human *TP53* function as GOF mutations to drive prostate tumorigenesis in mice and growth of human PCa cells in culture, respectively.

### Upregulation of pyrimidine synthesis genes (PSGs) by ERG and GOF mutant p53

To define the downstream effectors uniquely altered in *Pb-T2-ERG;Trp53*[pcR172H/-] but not *Pb-T2-ERG;Trp53*[pc/-] mice, we performed RNA-seq analysis in prostate tissue from the six groups of GEM shown in Fig. 1c. Gene clustering analysis of the RNA-seq data revealed that prostate tumors from *Pb-T2-ERG;Trp53*[pc/-] mice and *Pb-T2-ERG;Trp53*[pcR172H/-] PIN lesions shared 370 commonly upregulated genes, but had 901 and 304 uniquely upregulated targets, respectively (Fig. 2a and Supplementary Fig. 3a, b and Supplementary Data 1). Through integration analysis of this set of genes and the ERG ChIP-seq data generated from *TMPRSS2-ERG* prostate tumors in a previously reported GEM model[25], we identified 531 ERG target genes that were highly upregulated in prostate tumors from *Pb-T2-ERG;Trp53*[pcR172H/-] mice compared to other genotypic littermates (Fig. 2b, c and Supplementary Data 2). IPA analysis showed that most of these genes are cancer relevant (Fig. 2d). Notably, a subset of PSGs[40], including *Umps*, *Rrm1*, *Rrm2* and *Tyms*, were upregulated in *Pb-T2-ERG;Trp53*[pcR172H/-]

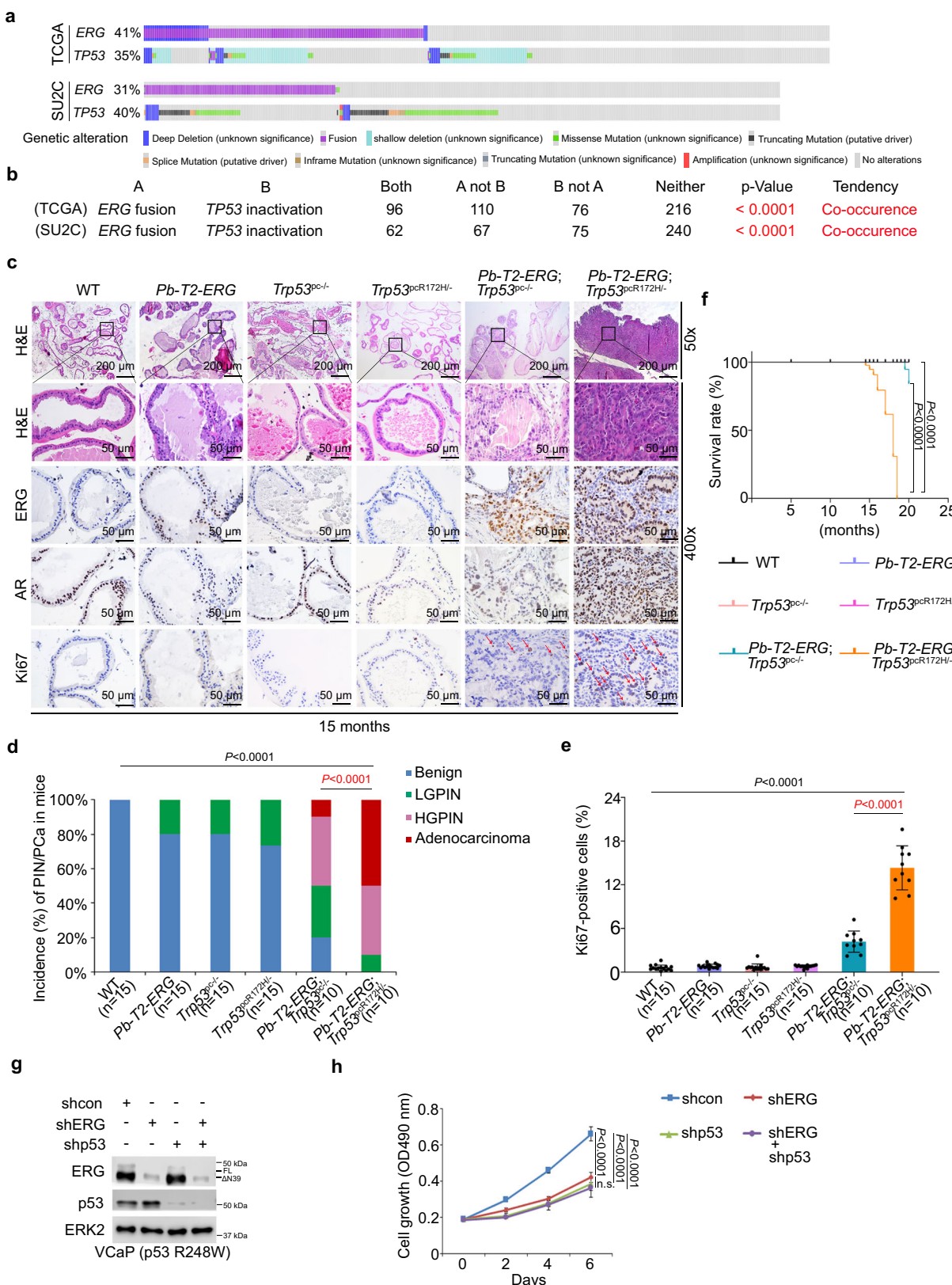

**Nature Communications** | (2023)14:4671

tumors compared to prostate tissues from control groups of mice (Fig. 2a, c–g and Supplementary Fig. 3c–e). These results were further validated by RT-qPCR in *Pb-T2-ERG;Trp53*[pcR172H/-] murine tumors (Fig. 2h, i). Similarly, expression of these PSGs was much higher in prostate tumors from the *Pten*[pc-/-]*;Trp53*[pcR172H/-]*;Pb-T2-ERG* GEM model we reported previously[41] compared to wild-type mice (Supplementary

Fig. 4a–e). In contrast, knockdown of p53 mutant, ERG or both decreased expression of these PSGs, but had no obvious effect on expression of nucleotide metabolism genes such as *DCK*, *TK1* and *IMPDH1* (Fig. 2i and Supplementary Fig. 4f, g). These data suggest that p53 GOF mutant cooperates with T2-ERG to upregulate expression of PSGs in PCa.

**Fig. 1 | *TMPRSS2-ERG* fusion and *TP53* inactivation alteration co-occur in PCa patient samples and cooperatively induce prostate tumorigenesis in mice.** **a** OncoPrint image from cBioPortal showing the percentage of genetic alterations in the *ERG* and *TP53* genes in PCa patients from TCGA (top) and SU2C (low) cohorts. **b** Fisher exact test (two-tailed) of the association between *TMPRRS2-ERG* fusions and *TP53* inactivation alterations in TCGA (top) and SU2C (low) PCa patient samples. **c** Representative images of H&E and IHC of ERG, AR and Ki67 proteins in prostate tissues from mice with the indicated genotypes at 15 months of age. **d** Quantification of incidences of PIN and/or cancer in mice with the indicated genotypes shown in (**c**). **e** Quantification of Ki67 positive cells from tissue sections

in (**c**). **f** The survival rate of mice with the indicated genotypes. **g** Western blot analysis of indicated proteins in VCaP cells stably expressing the indicated shRNAs. ERK2 was used as a loading control. The western blot assay was repeated two times independently with similar results. **h** MTS assay in VCaP cells stably expressing the indicated shRNAs. n.s., nonsignificant. Data in (**d, e**) and (**f**) represented as mean ± s.d. from indicated sample size (numbers of mice in each genotypic group). Data in (**h**) represented as mean ± s.d. from five independent replicates for each group. χ2 test was performed in (**d**) for statistical analysis. Two-tailed Student's *t* test was performed in (**e**). The Log-rank (Mantel-Cox) test was performed in (**f**). Two-way ANOVA was performed in (**h**).

## PSG expression is not directly regulated by p53 GOF mutant

To determine whether a p53 GOF mutant regulates expression of the PSGs by binding to their genomic loci, we performed p53 ChIP-seq in VCaP cells using anti-p53 antibody (DO-1). We identified 1116 peaks significantly bound by p53 GOF mutant R248W ($P < 1E-10$) and these peaks were localized in both promoter and non-promoter regions and are associated with 615 genes (Fig. 3a, Supplementary Fig. 5a and Supplementary Data 3). Consistent with the notion that the mutant p53 protein fails to bind to cognate WT p53 binding elements in the genome[10], we did not detect any obvious binding of p53 GOF mutant R248W at canonical p53 target genes in VCaP cells. Intriguingly, there was no R248W binding at the PSG loci either in VCaP cells (Supplementary Data 3), suggesting that p53 GOF mutant regulates PSG expression through indirect mechanism(s).

## Identification of *CTNNB1* as a transcription target gene of p53 GOF mutant

To define the downstream effector(s) mediating p53 GOF mutant regulation of PSG expression, we conducted Gene Ontology (GO) analysis of p53 mutant R248W-bound genes. We identified the transcriptional coregulator binding pathway as the top enriched pathway (Fig. 3b) and detected a R248W-bound peak in the promoter of *CTNNB1* gene which encodes β-Catenin, a core transcriptional component of the β-Catenin/TCF complexes[42] (Fig. 3c). This result was further validated by ChIP-qPCR in VCaP cells (Fig. 3d). Meta-analysis of p53 ChIP-seq data generated from breast cancer cell lines[16] showed that other p53 GOF mutants including R273H, R248Q and R249S, but not WT p53, also bound at the *CTNNB1* promoter (Supplementary Fig. 5b).

Next, we sought to determine the molecular mechanisms by which the p53 GOF mutant regulates the *CTNNB1* gene promoter. Previous studies have indicated that p53 DBD mutants promote different oncogenic transcriptional programs through protein interaction with other transcription factors such as ETS family member ETS2 in cell lines of different cancer types[14–16]. Results from co-IP and proximity ligation assay (PLA) suggested an interaction between ERG and p53 R248 mutant when expressed at endogenous levels in VCaP cells; however, the interaction disappeared when ethidium bromide was added to cell lysate used for co-IP (Supplementary Fig. 5c, d), suggesting an indirect interaction between ERG and p53 mutant R248W in VCaP cells. In agreement with this notion, in vitro protein binding assay using in vitro translated proteins showed that there was no interaction between ERG and any of p53 GOF mutants examined (Supplementary Fig. 4e). These data indicate that, in contrast to the mechanism of action for ETS2, it is less likely that p53 GOF mutant is recruited to the *CTNNB1* gene promoter through its interaction with ERG. Consistent with this observation, DNA binding motif analysis showed that no typical transcription factor binding motif was specifically enriched among R248W mutant-bound gene loci (Supplementary Fig. 4f).

Next, we sought to test the hypothesis that p53 GOF mutant regulates *CTNNB1* expression by directly binding to the DNA sequence in the *CTNNB1* promoter. To this end, we performed p53 R248W ChIP-qPCR analysis using a sequential set of primers (Fig. 3e). We

demonstrated that the R248W mutant specifically occupied the center (#b amplicon) of the ChIP-seq peak in VCaP cells (Fig. 3e, f). To define the mutant p53 binding sequence (MP53BS), we performed electrophoretic mobility shift assays (EMSA) using VCaP cell lysate and four biotin-labeled double-stranded DNA probes covering the #b amplicon region (Fig. 3e and Supplementary Fig. 5g). We identified a 25-bp MP53BS in the *CTNNB1* gene promoter (Fig. 3e, g). The EMSA signal of the MP53BS probe was substantially diminished by adding competing unlabeled probe or anti-p53 antibody in the assays (Fig. 3h, i), indicating that the detected binding signal is specific for the p53 R248W mutant. Besides using nuclear extract, we also purified from bacteria the glutathione S-transferase (GST)-p53 recombinant fusion proteins, including WT p53 and various mutants such as R175H (equivalent to R172H in GEM mice), C238Y (LuCaP 23.1 patient-derived xenograft (PDX)), R248W (VCaP cell line), R273H (MDA-MB-468 breast cancer cell line) and Q331R (22Rv1 PCa cells), a residue outside of DBD. We performed EMSA using untagged p53 recombinant proteins after cleavage of the GST tag. We confirmed that, except for WT and Q331R, all the DBD mutants of p53 bound to the MP53BS probe (Fig. 3j), suggesting that the DBD mutants of p53 can directly bind to the MP53BS derived from the *CTNNB1* gene promoter. The MP53BS was only bound by mutant p53 (R248W) in VCaP cell lysate but not WT p53 in LNCaP cell lysate and on the contrary, the WT p53 binding sequence from *CDKN1A* gene was only bound by WT p53 in LNCaP cell lysate, but mot mutant p53 (R248W) from VCaP cell lysate (Supplementary Fig. 4h). Notably, MP53BS in the *CTNNB1* promoter shares approximately 50% of homology with the WT p53 binding consensus sequence (Supplementary Fig. 4i), indicating that these sequences might be related. An almost identical motif is also observed in the mouse *Ctnnb1* promoter (Supplementary Fig. 4i). A similar C-rich motif can be found in the promoters of previously reported p53 GOF mutant-bound cancer-related genes such as *KMT2A* and *KAT6A*[16] (Supplementary Fig. 4i and Supplementary Data 4). Moreover, we also tested the MP53BS C-to-A mutants and demonstrated the importance of the C-rich sequence for mutant p53 binding to DNA (Supplementary Fig. 4j).

Similar to VCaP cells, DU145 cells express mutated p53 (R223L in one allele and V274F in the other) and a ETS fusion (ETV4)[3,43]. ChIP-qPCR analysis showed that mutant p53 also occupied the *CTNNB1* gene promoter in DU145 cells, and similar results were obtained from organoids from *Trp53*[pcR172H/-] mouse prostates (Supplementary Fig. 5j, k). Knockout of the p53 GOF mutant by CRISPR/Cas9 technology in DU145 cells not only abolished mutated p53 occupancy at the *CTNNB1* promoter, but also largely decreased *CTNNB1* mRNA and β-Catenin protein expression (Supplementary Fig. 5l–n). In contrast, endogenous WT p53 failed to bind to the *CTNNB1* promoter in LNCaP cells and knockout of WT p53 had no obvious effect on β-Catenin mRNA and protein expression in these cells (Supplementary Fig. 5j, 6a, b). These data suggest that GOF p53 mutants (e.g., R248W and R223L/ V274F) share the ability to bind the *CTNNB1* promoter and upregulate *CTNNB1* gene expression in PCa cells.

To further investigate the molecular mechanism by which GOF mutant p53 regulates *CTNNB1* gene transcription, we deleted the genomic region of the MP53BS in the *CTNNB1* promoter in DU145 cells using CRISPR/Cas9 technology with one pair of sgRNAs. Using both

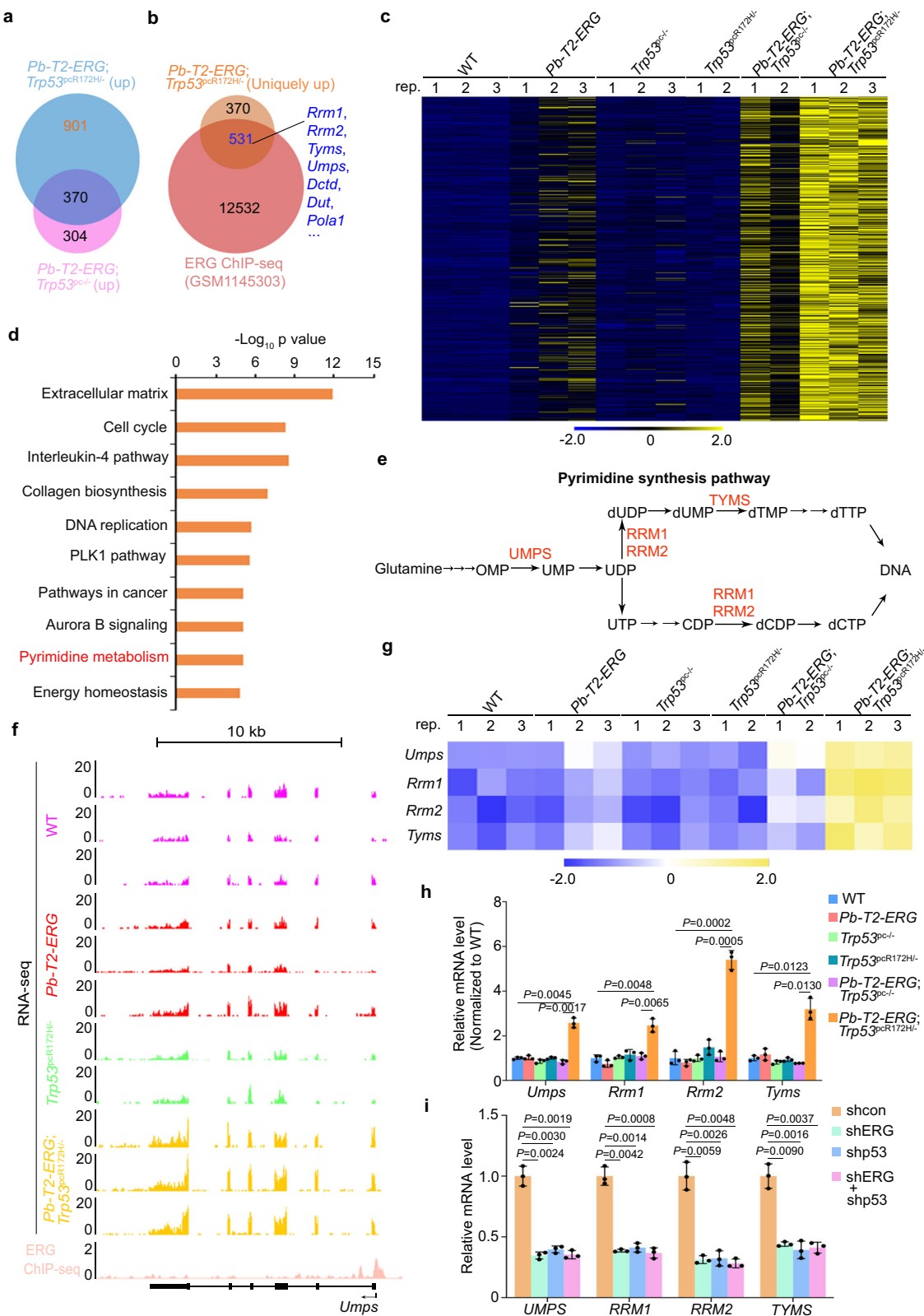

PCR and DNA Sanger sequencing approaches, we identified four MP53BS deletion clones #7, #17, #19 and #32 (Supplementary Fig. 6c). Notably, both *CTNNB1* mRNA and β-Catenin protein expression were largely downregulated in all four clones (Fig. 3k, l). While knockdown of the p53 GOF mutant markedly decreased β-Catenin mRNA and protein expression in control DU145 cells, it failed to further decrease β-Catenin expression in MP53BS KO cells (Fig. 3m, n). We carried out

additional EMSA using point mutants of MP53BS in which cytosine nucleotides in two cytosine-rich regions were mutated to adenine and found that mutation of these two regions also completely abolished mutant p53 binding of MP53BS (Supplementary Fig. 6d).

The *CTNNB1* mRNA expression level was much higher in p53-mutated PCa cell lines VCaP (p53 R248W) and DU145 (p53 R223L/V274F) compared to p53-WT or p53-null cell lines including C4-2,

**Fig. 2 | Co-regulation of pyrimidine synthesis genes by ERG and GOF mutant p53 in GEM prostate tumors and human PCa cell line. a** Venn diagram showing the overlap between the genes upregulated in prostate tumors of *Pb-T2-ERG;Trp53*[pcR172H/-] mice (*n* = 3, 15 months old) and those upregulated in prostatic tissues of *Pb-T2-ERG;Trp53*[pc-/-] mice (*n* = 3, 15 months old) revealed by RNA-seq data. **b** Venn diagram showing the overlap between the genes (*n* = 901) uniquely upregulated in prostate tumors of *Pb-T2-ERG;Trp53*[pcR172H/-] mice (*n* = 3, 15 months old) as defined in (**a**) and ERG-bound gene targets identified by ChIP-seq in murine prostate tumors (GSM1145303). **c** Heatmap of RNA-seq data showing a subset of genes (*n* = 531) differentially expressed in the benign or cancerous prostate tissues of mice (15 months old) with the indicated genotypes (*n* = 3 except the *Trp53*[pcR172H/-] single mutant group and *Pb-T2-ERG;Trp53*[pc-/-] group). **d** KEGG pathway analysis of 531 ERG target genes that were uniquely upregulated in prostate tumors of *Pb-T2-ERG;Trp53*[pcR172H/-] mice as shown in (**c**). **e** Diagram depicting the key enzymes including UMPS, RRM1, RRM2 and TYMS in pyrimidine synthesis. **f** UCSC Genome Browser screenshots in the *Umps* gene locus showing the results of RNA-seq data from the benign or cancerous prostate tissues of mice (15 months old) with the indicated genotypes (*n* = 3 except the *Trp53*[pcR172H/-] single mutant group) and the ERG ChIP-seq data from murine prostate tumors (GSM1145303). **g** Heatmap of RNA-seq data showing expression of the indicated PSGs in the benign or cancerous prostate tissues of mice (15 months old) with the indicated genotypes. **h** RT-qPCR analysis of expression of the indicated PSGs in the benign or cancerous prostate tissues of mice (15 months old) with the indicated genotypes. **i** RT-qPCR analysis of indicated proteins and PSG gene mRNAs in VCaP cells stably expressing control (shcon) or gene-specific shRNAs. Data in (**h**) and (**i**) were shown as mean ± s.d. from three independent experiments. Two-tailed Student's *t* test was performed in (**h**) and (**i**).

LNCaP, PC-3 and 22Rv1 (Supplementary Fig. 6e). Accordingly, knockdown of mutated p53, ERG or both attenuated expression of canonical β-Catenin target genes including *AXIN2*, *LEF1* and *c-MYC* in VCaP cells, and similar results were obtained by knockdown of the p53 mutant in DU145 cells (Supplementary Fig. 6f, g). In several non-prostate cancer types, we found that *CTNNB1* expression was also significantly higher in *TP53* mutated, but not *TP53*-null or WT breast and pancreatic cancer patient samples, but no correction was observed in bladder and colon cancer samples (Supplementary Fig. S6h). Taken together, these data suggest that β-Catenin is a regulatory target of p53 GOF mutants and regulation is mediated through the MP53BS in the *CTNNB1* gene promoter. However, upregulation of *CTNNB1* by mutated p53 is apparently influenced by other factors in a context-dependent manner.

## TMPRSS2-ERG cooperates with a p53 GOF mutant in regulating β-Catenin expression

Analysis of RNA-seq data showed that *Ctnnb1* mRNA was upregulated in prostate tumors of *Pb-T2-ERG;Trp53*[pcR172H/-] mice, but not in prostatic tissues of *Pb-T2-ERG* and *Trp53*[pcR172H/-] single mutant mice relative to WT mice, and this observation was further confirmed by RT-qPCR (Fig. 4a, b, c). Similarly, *CTNNB1* mRNA was only significantly upregulated in *ERG* fusion and *TP53* mutation dual-positive, but not single-positive patient samples of the SU2C cohort (Fig. 4d). Both mouse and patient data imply that ERG may cooperate with a p53 GOF mutant to regulate *CTNNB1* expression. By analyzing ERG ChIP-seq data in VCaP cells, we not only found that ERG bound to the *CTNNB1* gene promoter, but also identified a core element of the ERG binding sequence (ERGBS) adjacent to the MP53BS in this locus (Fig. 4e). ChIP and re-ChIP assay confirmed that mutant p53 and ERG both bound to the same region in *CTNNB1* promoter in VCaP cells (Fig. 4f). These data suggest that ERG and mutant p53 collaborate to regulate β-Catenin expression through co-occupancy of a site on the *CTNNB1* promoter. We further showed that ERG knockdown in VCaP cells also downregulated *CTNNB1* mRNA and β-Catenin protein level, albeit not as effectively as mutant p53 depletion (Fig. 4g, h), supporting a predominant role of GOF mutant p53 in regulation of β-Catenin expression in PCa cells. We also found that the enrichment of H3K27Ac (an active marker for gene transcription) at *CTNNB1* promoter was decreased by knockdown of either ERG or p53 mutant in VCaP cells (Fig. 4i). Luciferase reporter assay showed that *CTNNB1* promoter activity was increased by expression of different p53 GOF mutants individually, but not ERG alone; however, promoter activity was substantially enhanced by co-expression of p53 mutants and ERG (Fig. 4j). These data suggest that TMPRSS2-ERG cooperates with a p53 GOF mutant to regulate expression of *CTNNB1* in PCa cells.

## Coregulation of PSG expression by TMPRSS2-ERG and β-Catenin in PCa

We next sought to define the molecular mechanism by which p53 GOF mutant and TMPRSS2-ERG regulate PSG expression. Given that there was no detectable binding of p53 R248W GOF mutant at the genomic locus of any PSG in VCaP cells (Supplementary Data 3). We sought to determine whether β-Catenin binds to PSG genomic loci and regulates their expression. Similar to the occupancy of ERG, meta-analysis of β-Catenin ChIP-seq data[44] showed that β-Catenin also bound to the promoter and/or non-promoter regions of PSGs including *UMPS, RRM1, RRM2* and *TYMS*, which encode key enzymes required for pyrimidine synthesis[45–47] (Fig. 5a and Supplementary Fig. 7a, b). Individual occupancy of ERG and β-Catenin at either promoter or putative enhancer in these loci were further confirmed by ChIP-qPCR (Fig. 5b, c).

Similar to the effect of depletion of ERG or p53 R248W, single β-Catenin knockdown inhibited mRNA and protein expression of these PSGs in VCaP cells (Fig. 5d, e), suggesting an essential role of β-Catenin in regulating PSG expression in PCa cells. ERG or p53 R248W knockdown did not result in further reduction in their expression in β-Catenin-deficient cells (Fig. 5d, e). Considering that mutant p53 cannot bind to PSG loci (Supplementary Data 3), these data suggest that β-Catenin is an essential downstream effector of p53 GOF mutant that cooperates with ERG to induce PSG expression. This notion is further supported by our observation that ERG and β-Catenin co-occupied the same region in the *UMPS* gene promoter in VCaP cells (Supplementary Fig. 7c). Similarly, co-expression of TMPRSS2-ERG (ERGΔN39) and p53 mutant (R248W), but neither alone, substantially increased expression of *UMPS* and other PSGs at mRNA and protein levels in p53-KO DU145 cells (Supplementary Fig. 7d, e). Consistent with the ChIP-seq data showing that ERG and β-Catenin occupy different regions in the PSG loci such as *RRM1, RRM2* and *TYMS*, chromatin conformation capture (3C) assay showed that ectopically expressed ERGΔN39 and p53 GOF mutant (R248W) induced chromatin interaction between ERG-occupied promoter and β-Catenin-bound non-promoter (putative enhancer) regions in these loci (Supplementary Fig. 7f–h). However, the increased chromatin interaction and expression of these PSGs were completely reversed by β-Catenin knockdown (Supplementary Fig. 7f–h). These results are concordant with enhanced enrichment of histone H3 lysine 27 acetylation (H3K27ac) and serine-2 phosphorylated RNA polymerase II (Pol II S2-p) in these loci (Supplementary Fig. 7i, j). Collectively, these findings support a hypothetical model wherein transcriptional upregulation of β-Catenin induced by mutant p53 and ERG cooperates with ERG in enhancing H3K27ac level and Pol II recruitment in PSG loci and promoting PSG expression in PCa cells (Supplementary Fig. 7k). In agreement with this notion, we found that high expression of *CTNNB1* mRNA significantly corelated with high mRNA levels of PSGs in ERG fusion-positive PCa patient samples (Supplementary Fig. 7l–o).

Next, we determined the impact of ERG and mutant p53 expression on pyrimidine synthesis and metabolism. We knocked down endogenous ERGΔN39 and p53 mutant R248W in VCaP cells and measured the level of UMP, dTTP and dTDP, three key intermediates for pyrimidine synthesis (Fig. 2e). Moreover, among six different PCa

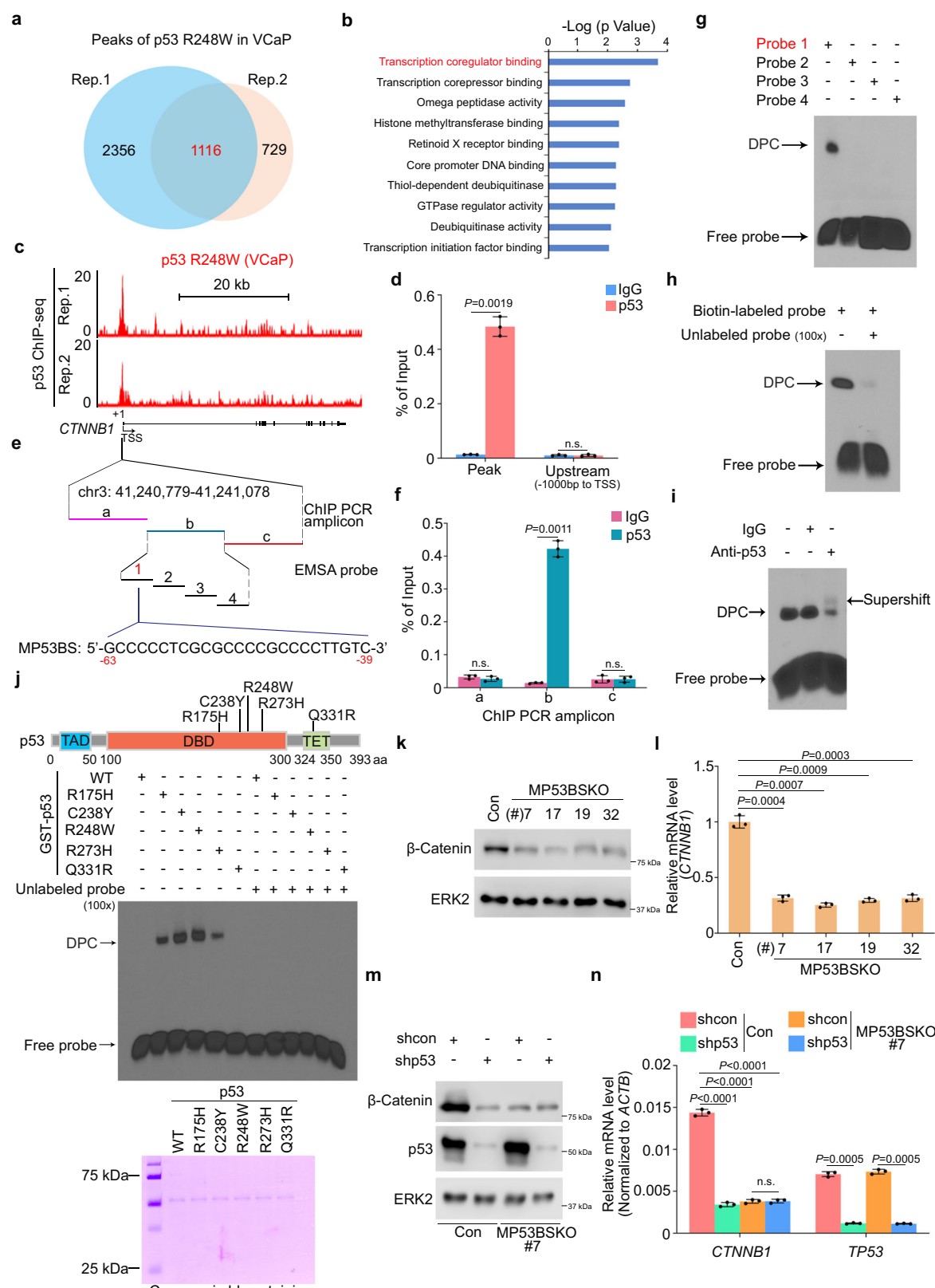

cell lines examined, PSGs were only highly expressed in the VCaP cell line, which is known to be positive for ERG fusion and p53 GOF mutation (Supplementary Fig. 8a). We further showed that co-knockdown of ERG and p53 R248W in VCaP cells significantly decreased the level of the pyrimidines and intermediates examined including UMP, dTDP and dTTP (Fig. 5f–g). Importantly, the supply of

pyrimidines partially rescued the growth of ERG- and p53 mutant-knockdown VCaP cells (Fig. 5h). These data indicate that ERG and p53 mutant are important for pyrimidine synthesis in TMPRSS2-ERG/ p53 mutant-positive PCa cells.

Since PSGs are important for nucleotide synthesis, we investigated whether expression of these PSGs play an important role in cell

**Fig. 3 | GOF mutant p53 binds to the promoter and transactivates expression of the *CTNNB1* gene. a** Venn diagram showing the 1116 mutant p53-bound peaks shared by two replicates (Rep.1 and Rep. 2) of p53 R248 ChIP-seq in VCaP cells. **b** Gene Ontology (GO) analysis of 615 p53-occupied target genes identified by ChIP-seq in VCaP cells. **c** UCSC Genome Browser screenshots showing the occupancy of mutant p53 R248W in the *CTNNB1* gene promoter in VCaP cells. **d** ChIP-qPCR analysis of mutant p53 R248W binding at the *CTNNB1* promoter in VCaP cells. n.s., not significant. **e** Scheme showing the locations of p53 ChIP-qPCR amplicons and EMSA DNA probes as well as the DNA sequence of MP53BS in the *CTNNB1* gene promoter. **f** ChIP-qPCR analysis of mutant p53 R248W binding at the *CTNNB1* promoter in VCaP cells using three sequential pairs of primers shown in (**e**). **g** EMSA assay using biotin-labeled double-stranded (ds) DNA probes from the *CTNNB1* promoter as indicated in (**e**) and nuclear extract from VCaP cells. DPC, DNA-protein complex. **h** EMSA assay using biotin-labeled and unlabeled ds DNA probe 1 as shown in (**e**) and nuclear extract from VCaP cells. **i** EMSA assay using biotin-labeled ds DNA probe 1 and nuclear extract from VCaP cells incubated with non-specific IgG (negative control) or anti-p53 antibody (DO-1). **j** Top, scheme showing the p53 missense mutants examined. Middle and Bottom, results of EMSA assay using biotin-labeled dsDNA probe 1 as shown in (**e**) or unlabeled probe and untagged p53 WT or indicated mutants purified from bacteria after GST cleavage. **k,l** Western blot (**k**) and RT-qPCR (**l**) analyses of indicated proteins and mRNAs in different clones of DU145 cells expressing MP53BS-targeting sgRNAs. **m,n** Western blot (**m**) and RT-qPCR (**n**) analyses of indicated proteins and mRNAs in control or the MP53BS KO #7 clone of DU145 cells expressing MP53BS-targeting sgRNAs. n.s. not significant. Data in (**d, f, l**) and (**n**) represented mean ± s.d. from three independent experiments. Both of the EMSA assays in (**g, h, i**) and (**j**), and the western blot assays in (**k**) and (**m**) were repeated two times independently with similar results. Data was performed by two-sided Fisher's exact test from DAVID Bioinformatics Resources (https://david.ncifcrf.gov/) in (**b**). Two-tailed Student's *t* test was performed in (**d, f, l**) and (**n**).

growth in TMPRSS2-ERG/p53 mutant double positive PCa. Individual or simultaneous depletion of UMPS, RRM1 and RRM2, three key enzymes for pyrimidine synthesis (Fig. 2e)[40] largely inhibited VCaP cell growth (Fig. 5i, j). On the contrary, we found that depletion of PSGs individually or together only slightly inhibited growth of ERG fusion- and p53 mutation-negative benign prostatic cell lines BPH1 and RWPE1 (Supplementary Fig. 8b–e), suggesting an essential role of increased expression of PSGs for growth of TMPRSS2-ERG/p53 mutant double positive PCa cells. We also examined the growth-inhibitory effect of PSG-targeting drugs such as hydroxyurea and gemcitabine (against RRM), 5-fluorouracil (against TYMS) and pyrazofurin (against UMPS) in VCaP cells. Similar to the previous reports in non-PCa cell types[48–50], low doses of gemcitabine were ineffective in inhibition of VCaP cell growth; however, growth of VCaP cells was effectively inhibited by high doses of gemcitabine (Supplementary Fig. 8f). Similar results were observed with high doses of hydroxyurea, 5-fluorouracil and pyrazofurin in VCaP cells (Supplementary Fig. 8g–i). These data suggest that existent drugs targeting the activity of the enzymes encoded by the PSGs such as could have therapeutic applications in TMPRSS2-ERG and GOF mutant p53 double positive PCa.

To investigate the role of the PSGs in vivo, VCaP cells expressing doxycycline (Dox)-inducible PSG shRNAs were inoculated into SCID mice to generate xenograft tumors. We demonstrated that Dox-induced PSG knockdown largely inhibited VCaP tumor growth in mice (Fig. 5k, l). Dox administration also decreased tumor weight, but had no obvious effect on mouse body weight (Fig. 5m and Supplementary Fig. 8j). These findings are further supported by the immunohistochemistry (IHC) data showing that expression of these three PSGs was largely knocked down and Ki67 level was substantially downregulated following Dox treatment (Supplementary Fig. 8k, l). Taken together, our data suggest that upregulation of key PSGs such as UMPS, RRM1 and RRM2 induced by ERG and p53 mutant is important for the growth of TMPRSS2-ERG and p53 mutant double positive PCa cells in vitro and in vivo.

### β-Catenin inhibitors suppress PSG expression and TMPRSS2-ERG/ p53 mutant positive PCa cell growth in vitro and in vivo

In agreement with the importance of β-Catenin in regulating PSG expression and the role of PSGs in promoting VCaP cell growth, we demonstrated that β-Catenin is also required for VCaP cell growth (Supplementary Fig. 9a, b), suggesting that β-Catenin could be a therapeutic target of TMPRSS2-ERG and p53 mutant-positive PCa. It has been reported that a small molecule inhibitor ICG-001 can inhibit the transcriptional activity of β-Catenin by binding to the transcription coactivator CREB-binding protein (CBP) and interrupting β-Catenin interaction with CBP[51]. PRI-724 is a pro-drug of C-82, another inhibitor of the β-Catenin-CBP interaction[52]. We therefore examined how sensitive the PCa cell lines, including VCaP (p53 mutant R248W), C4-2 (p53 WT), LNCaP (p53 WT), and PC-3 (p53 null), are to these two β-Catenin-

CBP interaction disruptors. We demonstrated that both drugs exhibited a greater growth-inhibitory effect in VCaP cells compared to other three cell lines (Fig. 6a, b), suggesting that TMPRSS2-ERG and p53 mutant-positive PCa cells are hyper-vulnerable to the inhibition of β-Catenin by these two compounds. We further showed that ICG-001 treatment of VCaP cells decreased expression of PSGs examined and canonical β-Catenin target genes *CCND1* and *c-MYC* at both mRNA and protein levels and inhibited cell growth in a dose-dependent manner (Fig. 6c–e). Similar results were obtained by treating VCaP cells with PRI-724 (Fig. 6f–h). In contrast, ICG-001 and PRI-724 treatment only inhibited the expression of *CCND1* and *c-MYC*, but not PSGs in ERG fusion- and p53 mutant-negative C4-2 and LNCaP cells (Supplementary Fig. 9c–j), further supporting the role of PSGs by β-Catenin in ERG-positive cells.

Next, we examined the anti-cancer effect of β-Catenin inhibitors in vivo. We chose ICG-001 since much higher concentrations of PRI-724 are required to achieve similar degrees of cell growth inhibition in vitro (Fig. 6a–h). We demonstrated that ICG-001 administration significantly inhibited growth of VCaP xenograft tumors in mice but had little or no effect on mouse body weight (Fig. 6i–k and Supplementary Fig. 9k). IHC analysis showed that ICG-001 treatment decreased the expression of pyrimidine synthesis enzymes such as UMPS and RRM1 and Ki67, but not CBP (Fig. 6l, m). Consistent with the inhibitory effect of ICG-001 on tumor growth and Ki67 expression, tumor growth returned when ICG-001 treatment was removed in the middle of treatment course (Supplementary Fig. 9l, m). Taken together, these results indicate that inhibition of the pyrimidine synthesis pathway by targeting the β-Catenin/CBP signaling nodule represents a viable therapeutic option to treat TMPRSS2-ERG/GOF p53 mutant-positive PCa.

### Therapeutic targeting of the β-Catenin-LEF/TCF complex in TMPRSS2-ERG/GOF mutant p53-positive PCa

β-Catenin transactivates its target genes by forming a protein complex with DNA binding partners LEF/TCF family proteins including LEF1, TCF1, TCF3 and TCF4[53]. We recently reported the development of oligonucleotide-based PROTACs (O'PROTACs or OPs) to target LEF1 for protein destruction[54]. As expected, an effective LEF1 O'PROTAC (OP-V1)[54] almost completely ablated LEF1 protein in VCaP cells. Of note, this O'PROTAC also downregulated TCF3 and TCF4 protein to a certain degree, consistent with the observation that members of the LEF/TCF protein family share a similar core DNA target sequence (e.g., TCAAAG) (Fig. 7a, b). TCF1 was not examined because its expression is low in VCaP cells, consistent with genotype-tissue expression (GTEx) RNA-seq data showing that TCF1 expression is undetectable in prostatic tissues. Importantly, this LEF1/TCF OP also inhibited protein expression of the key pyrimidine synthesis enzymes and VCaP cell growth in culture (Fig. 7b, c).

We further sought to determine the anti-cancer efficacy of LEF1/TCF O'PROTACs using ERG/GOF p53 mutant-positive PCa organoids

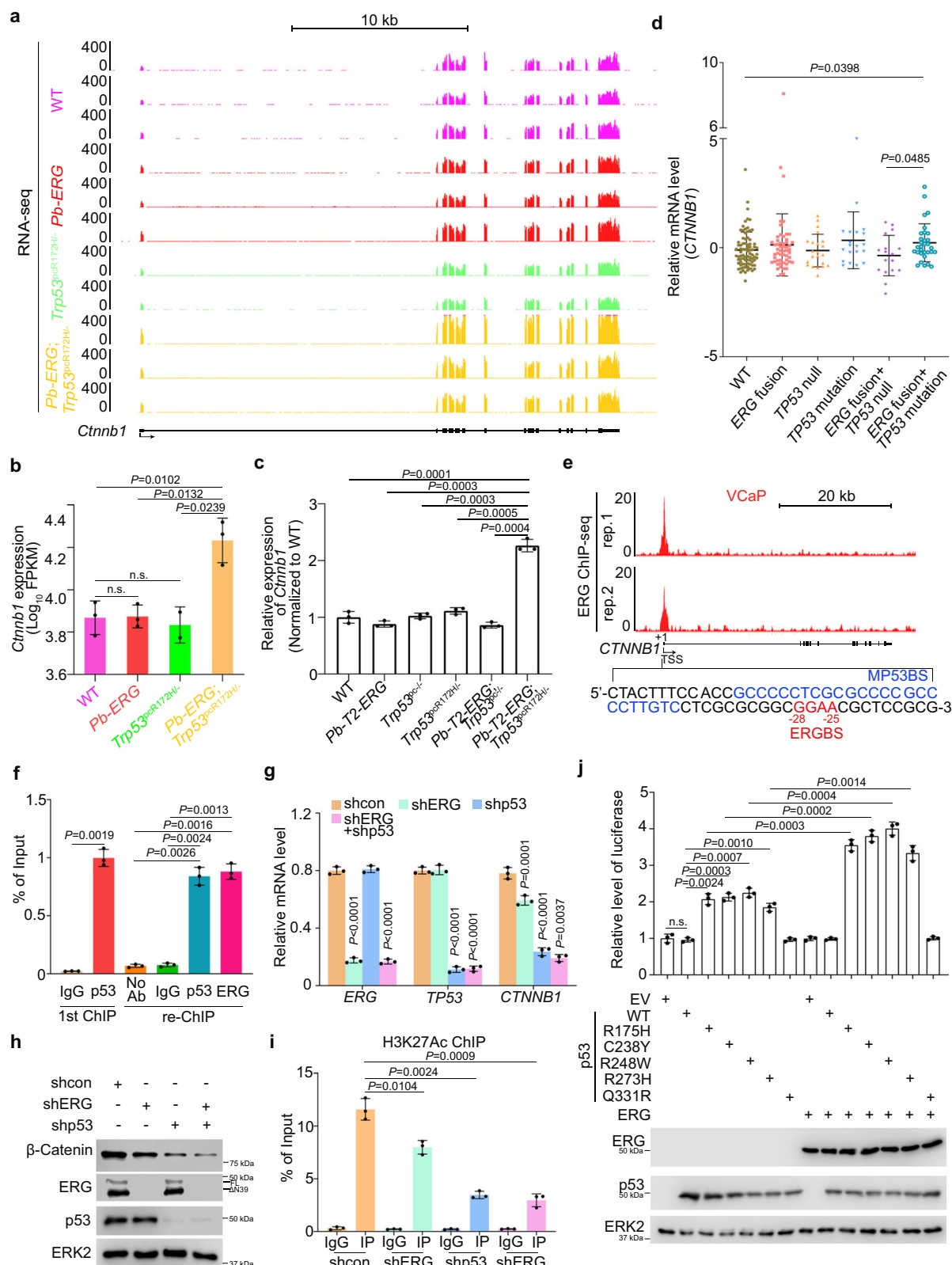

and PDXs. It has been reported that LuCaP 23.1 PDX and its androgen-independent (castration-resistant) subline LuCaP 23.1AI are *TMPRSS2-ERG* gene fusion positive and that one allele of *TP53* is deleted[55]. Consistent with the report in LuCaP 23.1AI[55], we confirmed that the parental LuCaP 23.1 PDX tumors also harbor a C238Y mutation in p53 DBD (Fig. 7d). In agreement with the EMSA result that mutant p53 C238Y bound to MP53BS in the *CTNNB1* promoter (Fig. 3j), knockdown

of this mutant by shRNAs largely decreased β-Catenin expression in LuCaP 23.1 PDX-derived organoids (PDXO) (Fig. 7e), indicating that LuCaP 23.1 is an ideal PDX model to test anti-cancer efficacy of inhibition of the β-Catenin-LEF/TCF pathway.

We demonstrated that LEF1/TCF OP treatment not only inhibited protein expression of key pyrimidine synthesis enzymes, but also effectively decreased growth of LuCaP 23.1 PDXO (Fig. 7f–h). Most

**Fig. 4 | ERG and GOF mutant p53 co-regulate β-Catenin expression. a** UCSC Genome Browser screenshots showing the *Ctnnb1* mRNA level revealed by RNA-seq in the benign or cancerous prostate tissues of mice (15 months old) with the indicated genotypes (*n* = 3 except the *Trp53*^pcR172H/- single mutant group). **b** Quantitative data showing the RNA-seq reads of *Ctnnb1* mRNA in the benign or cancerous prostate tissues of mice (15 months old) with the indicated genotypes (*n* = 3 except the *Trp53*^pcR172H/- single mutant group). Log₁₀ (FPKM) was calculated for the expression of *Ctnnb1* mRNA. **c** RT-qPCR analysis of *Ctnnb1* mRNA in the tissues from the mice with indicated genotypes. **d** Meta-analysis of RNA-seq data showing the *CTNNB1* mRNA expression levels in the indicated genotypic subgroups of patient samples from the SU2C cohort. **e** UCSC Genome Browser screenshots showing the occupancy of ERG in the *CTNNB1* gene promoter in VCaP cells. A putative ERG consensus binding motif is highlighted in red while MP53BS was highlighted in blue. **f** Sequential ChIP (Re-ChIP)-qPCR analysis of the co-binding of ERG and p53 mutant at the *CTNNB1* promoter in VCaP cells. (**g,h**) RT-qPCR (**g**) and

Western blot (**h**) analyses of indicated proteins and mRNAs in VCaP cells stably expressing the indicated shRNAs. **i** ChIP-qPCR analysis of H3K27Ac in the *CTNNB1* gene promoter in VCaP cells with the indicated shRNAs. **j** Top, Luciferase assay of *CTNNB1* promoter with indicated plasmids in DU145 p53KO cells. Bottom, western blot analysis of the indicated protein after transfection of indicated plasmids. Data in (**b**) and (**c**) represented mean ± s.d. from indicated sample size. Data in (**b**) represented mean ± s.d. from WT (*n* = 82), *ERG* fusion (*n* = 54), *TP53* null (*n* = 22), *TP53* mutation (*n* = 20), *ERG* fusion + *TP53* null (*n* = 17) and *ERG* fusion + *TP53* mutation (*n* = 31). Data in (**f, g, i**), and (**j**)were shown as mean ± s.d. from three independent experiments. The western blot assay in (**h**) was repeated two times independently with similar results. The western blot assays in (**j**) repeated three times independently with similar results. Two-sided Student's *t* test for the statistical analysis was performed in (**b, c, f, g**) and (**i**). Mann–Whitey *U* test was used in (**d**).

---

importantly, this effect was almost completely reversed by supplementation of dTTP/dCTP, but not dATP/dGTP deoxynucleoside triphosphates (Fig. 7g, h). On the contrary, the expression of PSGs was not changed significantly after the treatment of OP in SPOP Q165P PDXO which had no ERG alteration or p53 mutation (Supplementary Fig. 10a, b). Exogenous nucleotides did not promote the growth of the SPOP Q165P PDXO after the treatment of OP (Supplementary Fig. 10c), suggesting that the anti-cancer effect of LEF1/TCF OP is largely mediated through the inhibition of pyrimidine synthesis in TMPRSS2-ERG/GOF p53 mutant PCa. Compared to the effect of control OP or vehicle, treatment of LEF1/TCF OP markedly blocked growth of LuCaP 23.1 PDX tumors without causing any obvious reduction in body weight of mice (Fig. 7i–l). IHC and Western blot analyses showed that LEF1/TCF OP not only decreased the level of LEF/TCF proteins and pyrimidine synthesis enzymes such as UMPS and RRM1, but also largely reduced the number of Ki67-positive cells (Fig. 7m, n and Supplementary Fig. 10d). These results indicate that inhibition of β-Catenin and PSG expression by targeting LEF/TCF proteins using O'PROTAC can effectively block the growth of *TMPRSS2-ERG* fusion and GOF p53 mutant positive PCa in vitro and in vivo.

## Discussion

Studies in GEM models with different genetic backgrounds invariably show that ERG overexpression, which recapitulates the *TMPRSS2-ERG* fusion in patients, is insufficient to drive prostate oncogenesis in mice until very advanced age (>24 months)[25,28–31]. Further studies reveal that ERG overexpression requires combination with other lesions such as deletion of *PTEN* or *FOXO1* to drive tumorigenesis, although the underlying mechanism remains poorly understood[28,29,31]. Building on the observation of the co-occurrence of *TMPRSS2-ERG* fusion and *TP53* gene alterations (deletion and/or mutation) in patient samples, we generated previously uncharacterized ERG/p53 KO and ERG/p53 GOF mutant (R172H) KI GEM models. By analyzing these new mouse models we discovered that ERG overexpression cooperates with p53 inactivation (deletion) to drive prostate tumorigenesis. Moreover, these models reveal that expression of p53 GOF mutant R172H accelerates PCa progression in the presence of ERG-overexpression. We also identify *CTNNB1* as a transactivation target gene of a GOF mutant p53 and a promising function of this p53 mutant in upregulation of essential PSGs including *UMPS*, *RRM1*, *RR*M2 and *TYMS*.

Increased cell proliferation is a hallmark of cancer, which requires high-rate DNA replication and synthesis of nucleotides including pyrimidine[56]. Pyrimidines are critical for synthesis of both DNA and RNA and cell cycle progression[46,57] and essential for cancer cell self-renewal and proliferation[47,58–63]. We provide evidence that ERG and p53 GOF mutant play critical roles in pyrimidine synthesis by co-regulating expression of PSGs. Our study further unravels that the aberrantly activated pyrimidine synthesis pathway is a pivotal mechanism that

links the cooperativity of ERG and mutant p53 to cancer development and progression (Fig. 8).

Using both loss- and GOF approaches in PCa cell lines and/or GEM models we consistently showed that both ERG and GOF mutant p53 are required for PSG upregulation. While ERG binds the promoter of PSGs, ChIP-seq data indicate that there is no obvious GOF mutant p53 binding in these gene loci in cell lines of prostate and other cancer types. It has been reported previously that mutant p53 binds the promoters of a subset of nucleotide metabolism genes (NMGs)[14,64]. However, the data from those studies clearly show that mutant p53 occupancy at the promoters of PSGs such as *RRM1, RRM2* and *TYMS* was neglectable, i.e., approximately 10–100-fold lower after normalized by the binding of non-specific IgG in comparison to the binding level in the loci of other NMGs such as *DCK, TK1* and *IMPDH1*[14,64]. These results and those from our present study stress that GOF mutant p53 regulates PSG expression via indirect mechanism(s).

GOF of mutant p53 plays essential roles in driving cancer growth and progression[1,3]. While p53 mutations result in loss of the ability to bind the p53 consensus DNA sequence in chromatin, increasing evidence indicates that mutant p53 can gain functions to regulate gene expression by interacting with other transcriptional factors such as ETS2 and CREB1[14,15,65,66]. By analyzing ChIP-seq data generated in PCa cells and the published datasets in breast cancer cell lines[16] we now provide evidence that all the p53 mutants we examined including R248W, R273H, R249S and R248Q, but not WT p53, bind to the *CTNNB1* gene promoter. EMSA analysis showed that endogenous mutant p53 protein in PCa cell lysate binds to the unique MP53BS in the *CTNNB1* gene promoter. Using recombinant p53 mutant proteins purified from bacteria for EMSA, we further demonstrate that all the DBD mutants we examined, but not a mutant mapping outside the DBD, can directly bind to MP53BS. Additionally, we show that ectopic expression of GOF DBD mutants in the presence of overexpression of ERG (or other ETS proteins such as ETV4 in DU145 cells) induces expression of *CTNNB1* mRNA expression in both human and mouse PCa cells in culture and in mice. Most importantly, our data show that deletion of the MP53BS genomic region by CRISPR/Cas9 largely diminished *CTNNB1* gene expression in DU145 PCa cells and that no further reduction in *CTNNB1* expression following knockdown of GOF mutant p53. Thus, our data clearly show that *CTNNB1* is a bona fide transactivation target gene of GOF mutant p53 (Fig. 8).

GOF mutations in the p53 DBD can be categorized into contact mutations (e.g., R248 and R273) and conformational mutations (e.g., R175)[3]. We found that knockdown of all the DBD mutants examined, including R175H, R248W and R273H, invariably resulted in downregulation of β-Catenin expression. These findings were corroborated by the results obtained from other assays including ChIP-seq and EMSA using both nuclear extracts from cancer cell lines expressing different GOF p53 mutants and the recombinant proteins purified from bacteria. Thus, our data are consistent with the notion that while all are

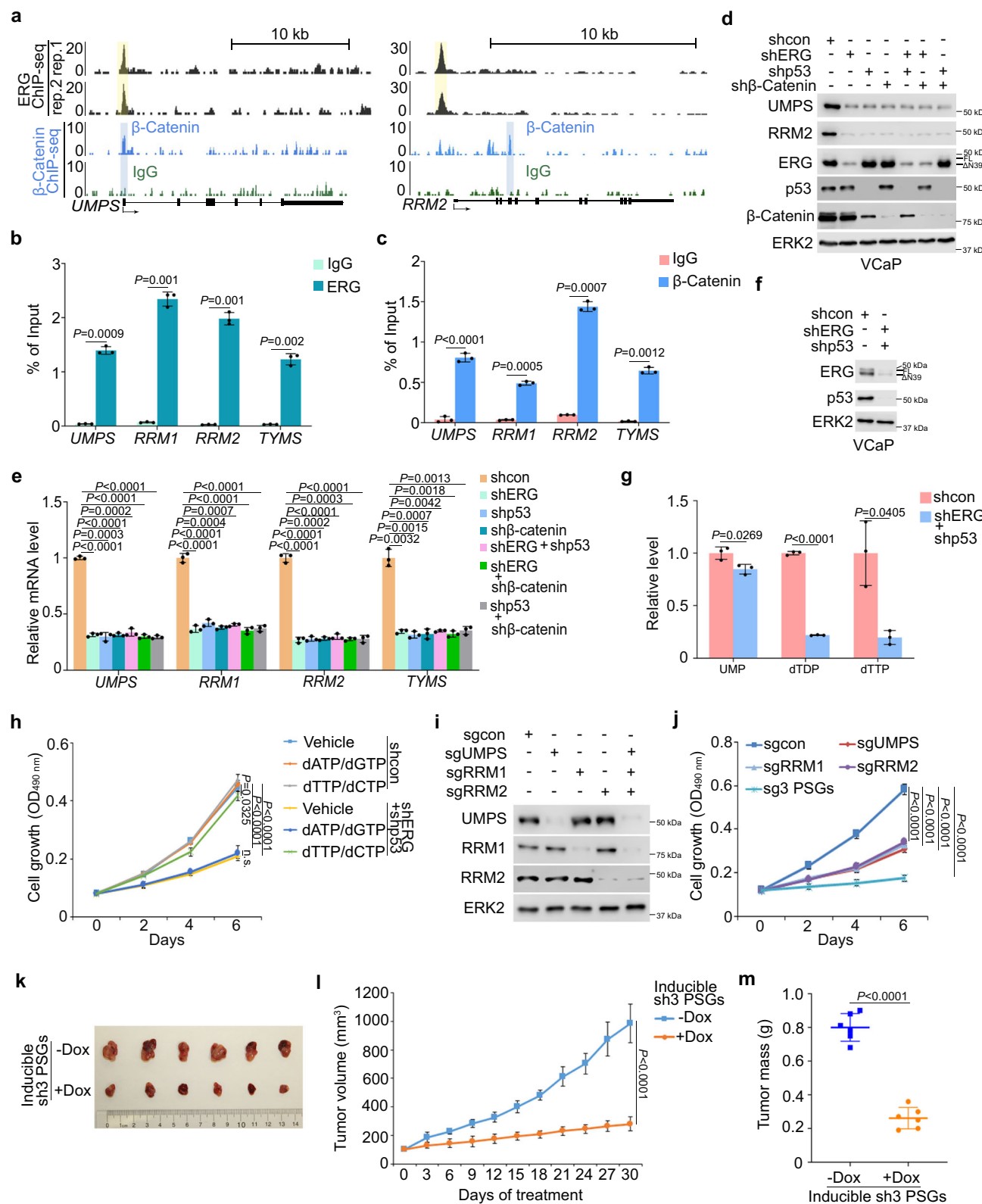

structurally different from WT p53, different mutation types (contact versus conformational) may all undergo certain shared conformation changes[3,5], allowing them to bind to a different set of DNA sequences. This notion is supported by our observation that while the MP53BS in *CTNNB1* gene promoter has only approximately 50% identity with the DNA consensus sequence of WT p53, a similar sequence can be evolutionally conserved in the mouse *Ctnnb1* gene promoter and present in a number of other GOF mutant p53 binding targets.

Our finding that β-Catenin plays a pivotal role in mediating PSG expression downstream of ERG and mutant p53 suggests that β-Catenin is a legitimate therapeutic target of ERG fusion- and p53 GOF mutant-positive PCa. Destruction of a target protein via PROTAC-induced proteolysis is emerging as a promising therapeutic strategy in cancer. To bypass the lack of small molecule inhibitors that bind β-Catenin, we developed a LEF1/TCF O'PROTAC strategy to block transcription activation by β-Catenin through targeted proteolysis. We

**Fig. 5 | ERG and β-Catenin co-occupy the PSG loci and co-regulate their expression. a** UCSC Genome Browser screenshots showing occupancy of ERG and β-Catenin in *UMPS* and *RRM2* gene loci revealed by ERG ChIP-seq in VCaP cells and β-Catenin ChIP-seq (GSE53927). **b,c** ChIP-qPCR analysis of occupancy of ERG (**b**) and β-Catenin (**c**) at *UMPS, RRM1, RRM2* and *TYMS* gene loci in VCaP cells. **d,e** Western blot (**d**) and RT-qPCR (**e**) analysis of indicated proteins and mRNAs in VCaP cells stably expressing indicated shRNAs. **f** Western blot analysis of indicated proteins in VCaP cells expressing indicated shRNAs. **g** Quantitative data showing the levels of UMP, dTDP and dTTP measured by LC-MS in VCaP cells generated as in (**f**). **h** MTS assay in VCaP cells with indicated shRNA as (**f**) and/or deoxynucleotides. **i** Western blot analysis of UMPS, RRM1 and RRM2 proteins in VCaP cells expressing indicated sgRNAs. **j** MTS assay in VCaP cells with depletion of indicated proteins as in (**i**). **k–m** VCaP cells expressing doxycycline (Dox)-inducible shRNAs for *UMPS, RRM1* and *RRM2* (sh3 PSGs) were inoculated into SCID mice and mice were treated with Dox daily for 30 days. Representative images of tumors harvested at the end of treatment (**k**), tumor growth curve (**l**) and tumor weight (**m**). Data in (**b, c, e**) and (**g**) were shown as mean ± s.d. from three independent experiments. The western blot assays in (**d, f**) and (**i**) were repeated two independent times with similar results. Data in (**h**) and (**j**) were shown as mean ± s.d. from five replicates. Data in (**l**) and (**m**) were shown as mean ± s.d. from six xenografts. Two-tailed Student's *t* test was performed in (**b, c, e, g**) and (**m**). Two-way ANOVA was performed in (**h, j**) and (**l**).

provide evidence that an oligonucleotide-based PROTAC is effective in inhibiting β-Catenin target gene expression and PCa cell growth in vitro and in vivo. Thus, our findings nominate β-Catenin as a promising therapeutic target of ERG/GOF mutant p53 dual positive PCa (Fig. 8).

In summary, we reveal an in vivo GOF role of mutant p53 in prostate oncogenesis. We identify a MP53BS in the *CTNNB1* gene promoter and demonstrate a DNA binding and transactivation function of GOF mutant p53 in transcriptional upregulation of *CTNNB1* oncogene. We further show that aberrantly overexpressed ERG cooperates with β-Catenin to transcriptionally upregulate PSGs by enhancing chromatin looping at these gene loci and promote ERG/GOF mutant p53-positive PCa growth via enhanced pyrimidine synthesis. Notably, LEF1 has been identified as a direct target of ERG[67]. These data imply that ERG regulates PSGs at different levels. Finally, we provide evidence suggesting that inhibition of β-Catenin by either small molecule inhibitors ICG-001 or PRI-724 or a LEF1/TCF oligonucleotide-based PROTAC represents a viable strategy not only suitable for effective treatment of ERG/GOF mutant p53-positive PCa, but also possibly for other cancer types such as the hematologic malignancies and solid tumors expressing GOF mutant p53 protein. Future DNase I footprinting studies are important for further assessment of the binding of MP53BS by p53 mutants, which is a limitation of the current study.

## Methods

### Antibodies, plasmids and chemicals

Antibodies used in the study include anti-ERG (1:1000 in dilution, BioCare, CM421C), anti-ERG (1:5000 in dilution, Abcam, ab92513), anti-p53 (1:1000 in dilution, Santa Cruz Biotechnology, sc-126), anti-ERK2 (1:1000 in dilution, Santa Cruz, sc-1647), anti-c-Myc (1:1000 in dilution, Santa Cruz Biotechnology, sc-40), anti-cyclin D1 (1:500 in dilution, Santa Cruz Biotechnology, sc-753), anti-CK8/CK18 (1:3000 in dilution, DSHB, AB 531826), anti-SMA (1:2000 in dilution, Dako, M0851), anti-active-β-Catenin (1:1000 in dilution, Millipore, 05-665), anti-β-Catenin (1:1000 in dilution, BD Biosciences, 610153), anti-RRM1 (1:1000 in dilution, Cell Signaling Technology, #8637), anti-RRM2 (1:1000 in dilution, Cell Signaling Technology, #65939), anti-UMPS (1:1000 in dilution, NOVUS, #85896), anti-AR (1:10,000 in dilution, Abcam, ab108341), anti-Ki67 (1:10,000 in dilution, Abcam, ab15580), anti-CBP (1:2000 in dilution, Santa Cruz Biotechnology, sc-583), anti-LEF1 (1:1000 in dilution, Cell Signaling Technology, #2230 S), anti-TCF3 (1:2000 in dilution, Proteintech, 14519-1-AP), anti-TCF4 (1:2000 in dilution, Proteintech, 22337-1-AP), anti-Histone H3 (acetyl K27) (2 μg for ChIP, Abcam, ab177178) and anti-RNA polymerase II CTD repeat YSPTSPS (phospho S2) (2 μg for ChIP, Abcam, ab5095). Anti-mouse secondary antibody (115-035-003, Jackson Immunoresearch), anti-mouse secondary antibody, light chain specific (115-035-174, Jackson Immunoresearch), anti-rabbit secondary antibody (111-035-144, Jackson Immunoresearch) at 1:5000 of dilution. Biotinylated IgG for IHC assay was purchased from Vector laboratory. shRNAs targeting *ERG* and *TP53* (p53) were purchased from Sigma-Aldrich and shRNAs targeting *CTNNB1* (β-Catenin) were acquired from the Mayo Clinic

RNA Interference Shared Resource (RISR). Mammalian expressing plasmids Tsin-SFB-p53 and Tsin-HA-ERG-ΔN39 were generated in house. The small guide RNA (sgRNA) sequences for CRISPR-Cas9-mediated gene deletion were designed using an online tool (https://portals.broadinstitute.org/gppx/crispick/public). The shRNAs targeting *RRM1, RRM2,* and *UMPS* genes were cloned into a doxycycline-inducible vector pINDUCER10[68] using XhoI and EcoR I cloning sites. The sequences of shRNAs and sgRNAs used are shown in Supplementary Data 5. ICG-001 (S2662) and PRI-724 (S8968) were purchased from Selleck. ICG-001 and PRI-724 were dissolved in DMSO (Sigma, D8418). LEF/TCF O'PROTACs were generated as we reported previously[54].

### Cell lines, organoids and their culture

VCaP, DU145, LNCaP, C4-2, PC-3, 22Rv1 and 293T cell lines were purchased from American Type Culture Collection (ATCC). DU145, LNCaP, C4-2, PC-3 and 22Rv1 cells were cultivated in RPMI 1640 media (Corning) supplemented with 10% fetal bovine serum (FBS) (Thermo Fisher Scientific). VCaP and 293T cells were grown in DMEM media (Corning) supplemented with 10% FBS (Thermo Fisher Scientific). All the cells were incubated at 37 °C supplied with 5% $CO_2$. Cells were treated with plasmocin (Invivogene) to eliminate mycoplasma prior to the subsequent experiments.

Organoids were generated from LuCaP 23.1 and SPOP Q165P patient-derived xenografts (PDXs) using the methods as described[69]. Organoids were cultured in FBS-free DMEM/F-12 medium mixed with Matrigel (Sigma) and other growth factors.

### Cell transfection and lentivirus infection

Cells were transiently transfected with indicated plasmids using either Lipofectamine 2000 (Thermo Fisher Scientific) or polyethylenimine (PEI) (PolySciences, Cat# 23966) according to the manufactures' instructions. For lentivirus package, 293T cells were co-transfected with plasmids for psPAX2, pMDG.2 and shRNAs using Lipofectamine 2000. Supernatant containing virus was harvested after 48 h and added into cells after filtration using 0.45 μm filter (Millipore). The virus-containing supernatant in the presence of polybrene (5 μg/ml) (Millipore) was added to the culture medium of target cells and infected cells were selected with 1 μg/ml puromycin (Selleck).

### Cell growth assay

Cells were seeded at the density of 1–5000 cells per well (depend on cell types) in 96-well plates overnight. At the indicated time points, optical density (OD) of cells was measured using a microplate reader (Biotek) at 490 nanometer after incubation with MTS solution (Promega) for 2 h at 37 °C in a cell incubator. For drug treatment, cells were seeded in 96-well plates overnight followed by adding the indicated compound(s) to each well. OD values were measured at the indicated time points using a microplate reader. For the O'PROTAC treatment, cells or organoids were transfected with the O'PROTAC compound at final concentration of 100 nM by using lipofectamine 2000 according to the manufacture's instruction. The nucleotides rescue experiments was performed as described with modifications[70]. Specifically, nucleotides at the final

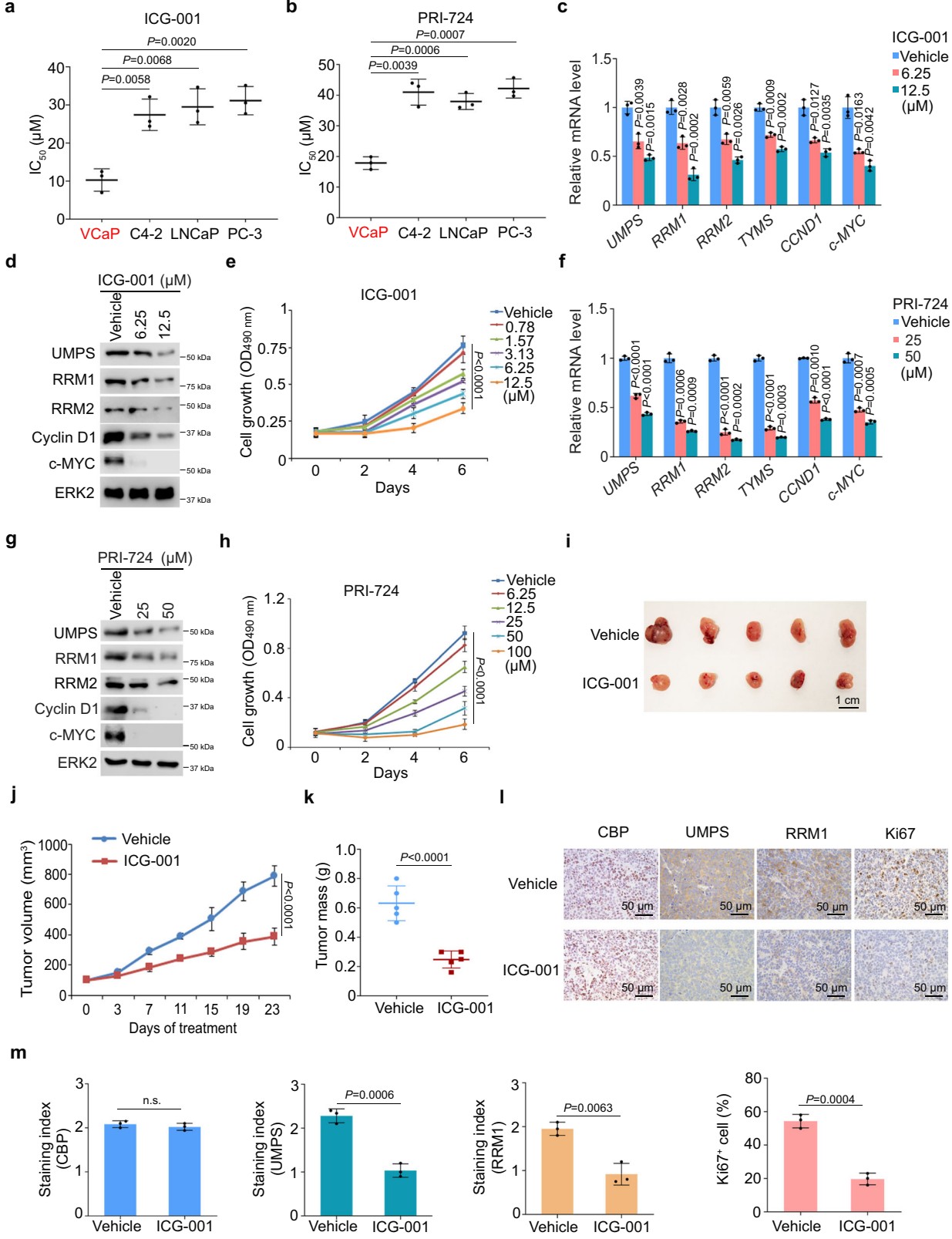

concentration of 10 μM by lipofectamine 2000 were mixed with O'PROTAC which was transfected into cells or organoids.

### Genetically engineered mouse model and genotyping

Mice were housed in 22 °C, 55% humidity on average with a 12-h light/12-h dark cycle and access to food and water. The indicated groups of target and control mice were bred by crossing mouse lines with the following genotypes: Probasin (*Pb*)-driven Cre4 recombinase transgenic mice, acquired from the National Cancer Institute (NCI) Mouse Repository and originally generated in the laboratory of Dr. Pradip Roy-Burman at University of Southern California (Los Angeles, CA)[37]; transgenic *ERG* mice purchased from the Jackson Laboratory (Cat# 010929), originally generated in the laboratory of Dr. Valeri Vasioukhin at Fred Hutchinson Cancer Research Center (Seattle, WA)[38]; *Trp53*

**Fig. 6 | β-Catenin inhibitors inhibit PSG expression and PCa xenograft growth in mice. a,b** IC$_{50}$ values of PCa cell lines treated with ICG-001 (**a**) or PRI-724 (**b**). **c,d** RT-qPCR (**c**) and Western blot (**d**) analysis of expression of indicated mRNAs and proteins in VCaP cells treated with vehicle or different doses of ICG-001. **e** MTS assay in VCaP cells treated with vehicle or different doses of ICG-001. **f,g** RT-qPCR (**f**) and Western blot (**g**) analysis of expression of indicated mRNAs and proteins in VCaP cells treated with vehicle or different doses of PRI-724. **h** MTS assay in VCaP cells treated with vehicle or different doses of PRI-724. **i–k** Representative images of VCaP xenograft tumors harvested after 23 days of ICG-001 treatment (**i**), tumor growth curve (**j**) and weight of tumors at day 23 of treatment (**k**). **l** Representative IHC images of indicated proteins in tumors as shown in (**i**). **m** Quantitative data of IHC intensity of each protein. n.s., not significant. Data in (**a**, **b**, **c**), and (**f**) were shown as means ± s.d. from three independent experiments. The western blot assays in (**d**) and (**g**) were repeated two independent times with similar results. Data in (**e**) and (**h**) were shown as means ± s.d. from five replicates. Data in (**j**) and (**k**) were shown as means ± s.d. from five xenografts. Data in (**m**) was shown as means ± s.d. from three independent experiments. For each experiment, five independent fields were enrolled for the calculation. Two-tailed Student's *t* test was performed in (**a**, **b**, **c**, **f**, **k**) and (**m**). Two-way ANOVA was performed in (**e**, **h**) and (**j**).

loxp/loxp conditional mice, acquired from the NCI Mouse Repository and originally generated in the laboratory of Dr. Tyler Jacks at Massachusetts Institute of Technology (Cambridge, MA)[12]; and *Trp53* loxp-STOP-loxp-R172H conditional mice, acquired from the NCI Mouse Repository and originally generated in the laboratory of Dr. Tyler Jacks[12]. PCR genotyping primers are listed in Supplementary Data 5. All animal studies were approved by the Mayo Clinic Institutional Animal Care and Use Committee (IACUC).

### Hematoxylin and eosin (H&E) staining and immunohistochemistry (IHC)
Four-μm sections were cut consecutively from formalin-fixed paraffin-embedded (FFPE) prostate tissues from GEM mice or xenograft or PDX tumors. Tissues were deparaffinized by xylene and subsequently rehydrated in turn through 100%, 95%, and 70% ethanal and water. After staining with hematoxylin and washing with Scott's Bluing solution (160 mM MgSO$_4$–7 H$_2$O, 25 mM sodium hydrogen carbonate), tissues were counterstained with 1% eosin. After washing with 95% ethanol, tissues were dehydrated with 95% and 100% ethanol. Finally, the stained tissues were treated with xylene and mounted with coverslips.

For IHC, sectioned tissues were rehydrated, endogenous peroxidase was inactivated and antigen retrieval was performed as previously described[71]. Antibodies for IHC as following: anti-AR (1:10,000 in dilution, ab108341, Abcam), anti-ERG (1:5000 in dilution, ab92513, Abcam), anti-Ki67 (1:10,000 in dilution, ab15580), anti-CK8/CK18 (1:3000 in dilution, Abcam, ab531826), anti-SMA (1:2000 in dilution, Dako, M0851), anti-active-β-Catenin (1:1000 in dilution, Millipore, 05-665), anti-UMPS (1:200 in dilution, NOVUS, #85896), anti-RRM1 (1:200 in dilution, Cell Signaling Technology, #8637), anti-CBP (1:100 in dilution, Santa Cruz Biotechnology, sc-583), anti-LEF1 (1:200 in dilution, Cell Signaling Technology, #2230 S). For quantification, the staining score was determined by multiplying the percentage of positive cells and the intensity ranged from 1 (weak staining), 2 (median staining), and 3 (strong staining). For Ki67 quantification, cells with positive staining in the nucleus were included to calculate the percentage of Ki67 positive-staining cells.

### RNA extraction and RT-qPCR
The total RNA was extracted from cultured cells or organoids using Trizol reagent (Thermo Fisher Scientific) according to the manufacturer's instructions. Complementary DNA was synthesized using reverse transcriptase (Promega). mRNA expression level was determined by real-time quantitative PCR (qPCR) using SYBR Green Mix (Thermo Fisher Scientific) with the realtime PCR system (Bio-Rad). Relative gene expression was normalized to the expression of housekeeping gene Actin Beta (*ACTB*). Primer sequences used for qPCR are listed in Table S5.

### Co-immunoprecipitation (co-IP) assay
Cells were collected and washed with cold 1 x PBS. Cells were lysed in IP buffer (0.5% NP-40, 20 mM Tris-HCl, pH = 8.0, 10 mM NaCl, 1 mM EDTA) supplemented with protease inhibitors (Sigma-Aldrich). Proteins from in vitro translation were synthesized by using TNT ® Quick Couple Transcription/Translation System (L1170, Promega). For the ethidium bromide (EtBr) treatment, cell lysate was incubated with 50 μg/ml EtBr for 30 min at 4 °C prior to immunoprecipitation. Anti-ERG or anti-p53 antibodies (2 μg) were added into cell lysate and incubated with Protein A/G beads (Millipore) overnight. Beads were washed and boiled with protein loading dye (Bio-Rad) for further analysis by Western blot.

### Proximity ligation assay (PLA)
VCaP cells were seeded into 6-well chamber slides before harvest. Cells were fixed with 4% paraformaldehyde followed by permeabilization in 0.4% Triton X-100. Cells were blocked in Duolink Blocking buffer (Sigma) for 1 h at 37 °C before in situ PLA assay. The PLA assay was performed according to the instruction of the Duolink in situ Red kit (Sigma-Aldrich, 92101). Primary antibodies with anti-ERG (1:200 in dilution) and anti-p53 (1:100 in dilution) were incubated overnight at 4 °C. The next day, Plus and Minus PLA probes were incubated for 1 h at 37 °C. Ligation and amplification of the PLA were performed. After several washes, cells were mounted in VECTASHIELD mounting medium with DAPI. Images were taken by using a confocal microscope (LSM880, Zeiss) with a 100x/1.3 Oil Objective.

### CRISPR/Cas9-mediated deletion of the MP53BS sequence in the *CTNNB1* gene promoter
A pair of sgRNAs for deletion of MP53BS in the *CTNNB1* gene promoter were designed, synthesized, and cloned into LentiCRISPR v2-dCas9 plasmid (Addgene, #112233). The generated plasmids were used for the lentivirus package and the infection of DU145 cells. At 48 h after infection, cells were cultured in fresh medium and selected with puromycin at a final concentration of 1 μg/ml. The parental DU145 cells and the selected clones were used for genomic DNA extraction for PCR amplification. PCR products were cloned into T vector (Takara, #3270) and subjected to Sanger sequencing. Primers of sgMP53BSKO are shown in Supplementary Data 5.

### Purification of GST-tagged recombinant proteins from bacteria
GST-tagged p53 expression plasmids for wild type (WT) and mutated p53 were transduced into *E. coli* BL21. The successfully transformed BL21 cells were cultured in flasks in an incubator shaker and treated with 100 μM IPTG (Sigma-Aldrich) at 18 °C overnight. The induced BL21 were collected and resuspended in lysis buffer (50 mM Tris-HCl, pH 8.0) containing protease inhibitors (Sigma-Aldrich) and sonicated. Glutathione agarose beads (Thermo Fisher Scientific) were added to pull down the GST-p53 (WT and mutant) recombinant proteins. Beads containing the recombinant proteins (1 μg) were incubated with 1 U Thrombin (Sigma-Aldrich, T4648) in PBS at room temperature for 8 h. The Thrombin was removed by incubated with Sepharose 6B beads (Bioworld, 20181111-2). The eluted protein in PBS was collected by centrifuge and saved for electrophoretic mobility shift assay (EMSA).

### Nuclear extract preparation and EMSA
Double-stranded DNA oligonucleotides were labeled with biotin as probes by using the commercial kit (Thermo Fisher Scientific, Cat# 89818). The labeled probes were incubated with nuclear extraction prepared from VCaP cells using NE-PER™ Nuclear and Cytoplasmic Extraction Reagents (Thermo Fisher Scientific, Cat# 78833) or purified

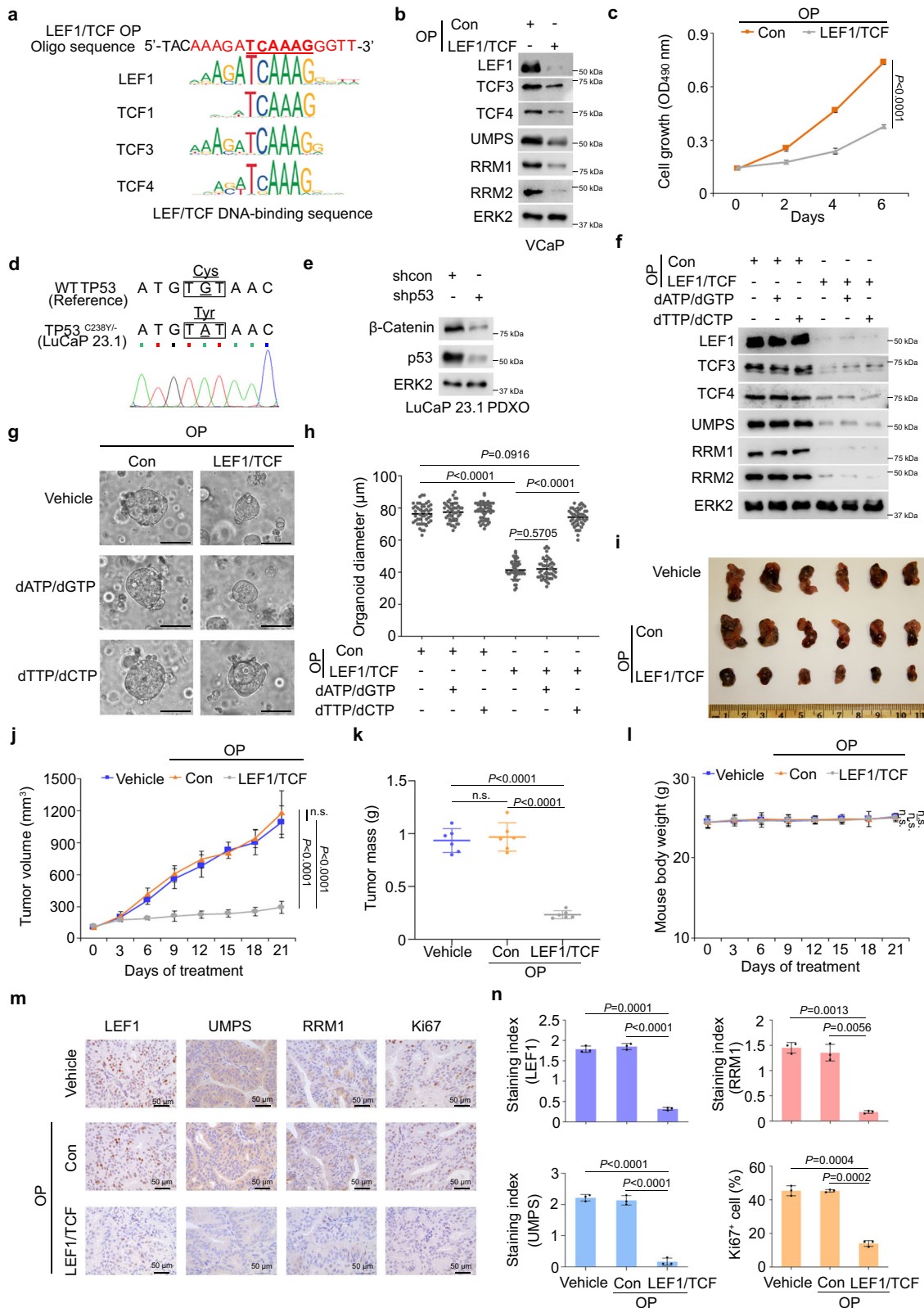

GST-p53 recombinant proteins according to the protocol provided by the manufacture (Thermo Fisher Scientific, Cat# 20148). For supershift assay, anti-p53 antibodies (1 μg) were added into the cell nuclear extract mixed with the biotin-labeled probes followed by incubation for 1 h at room temperature before loaded into 6% non-denatured polyacrylamide gel. DNA sequences of the probes are listed in Supplementary Data 5.

### RNA sequencing (RNA-seq) and data analysis

Prostate tissues from mice were dissected and collected for RNA extraction using the RNeasy Plus Mini Kit (Qiagen). The extracted RNA was subjected to the high throughput sequencing at the Mayo Clinic Genome Core Facilities. High quality total RNA with RNA integrity number (RIN) > 9.0 was used to generate the RNA-seq library using the TruSeq RNA Sample Prep Kit v2 (Illumina). RNA samples from

**Fig. 7 | LEF1/TCF O'PROTAC inhibits PSG expression and growth of TMPRSS2-ERG and mutant p53-positive PCa PDX tumors. a** Sequence alignment between the DNA oligonucleotide used in LEF1/TCF O'PROTAC (OP) and the consensus DNA binding elements of each member of the LEF/TCF family. **b** Western blot analysis of indicated proteins in VCaP cells treated with control or LEF1/TCF OP (100 nM) for 48 h. **c** MTS assay in VCaP cells treated with control or LEF1/TCF OP (100 nM). **d** Sanger sequencing confirmation of C238Y mutation in LuCaP 23.1 PDX tumor samples. **e** Western blot analysis of indicated proteins in organoids derived from LuCaP 23.1 PDXs (PDXO). **f–h** LuCaP 23.1 PDXOs were treated with indicated OP (100 nM) and/or deoxynucleotides and harvested for Western blot analysis at 48 h after treatment (**f**) or cultured for 3 days followed by photographing (**g**) and quantification of the diameters of organoids (**h**). n.s., not significant. **i–k** Representative images of LuCaP 23.1 PDX tumors at 21 days after OP treatment (10 mg/kg every other day) (**i**), growth curve of tumors over the 21-day treatment period (**j**)

and weight of tumors at day 21 of OP treatment (**k**). n.s., not significant. **l** Body weight of mice at 21 days after OP treatment as in (**j**). n.s., not significant. **m, n** Representative IHC images of the indicated proteins from tumors shown in (**i**) and quantification of IHC staining of the indicated proteins. See details of staining scoring and index in Methods. The western blot assays in (**b, e**) and (**f**) were repeated two independent times with similar results. Data in (**c**) was shown as mean ± s.d. from five replicates. Data in (**h**) was shown as mean ± s.d. from a total of 45 organoids randomly obtained from three independent experiments. Data in (**j**) and (**k**) were shown as mean ± s.d. from six PDXs. Data in **l** was shown as mean ± s.d. from six mice. Data in (**n**) was shown as mean ± s.d. from three independent experiments. For each experiment, five independent fields were enrolled for the calculation. Two-way ANOVA was performed in (**c, j**) and (**l**). Two-tailed Student's *t* test was performed in (**h, k**) and (**n**).

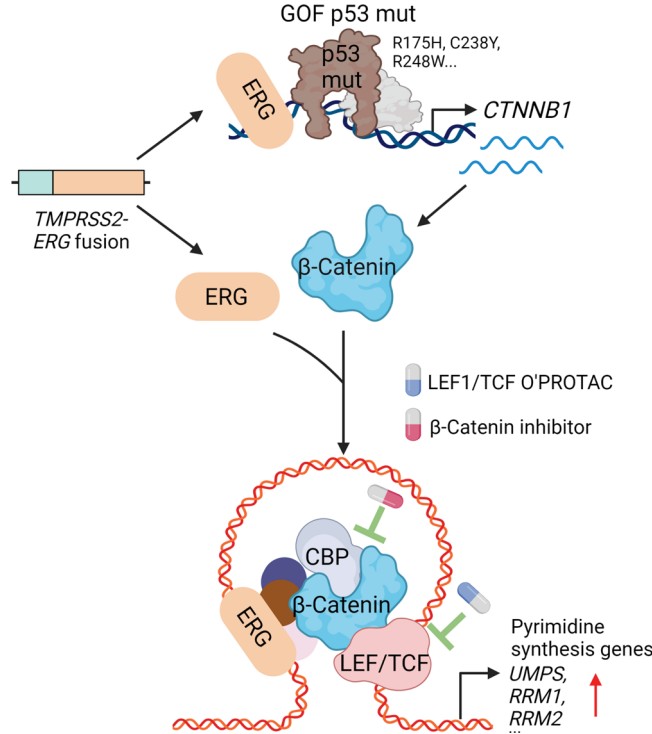

**Fig. 8 | A hypothetical model deciphering the cooperativity of TMPRSS2-ERG and GOF mutant p53 in prostate oncogenesis and progression.** GOF mutant p53 binds to the *CTNNB1* gene promoter and transactivates *CTNNB1* expression by cooperating with overexpressed ERG caused by *TMPRSS2-ERG* fusion. Increased expression of β-Catenin induces pyrimidine synthesis gene (PSG) expression by interacting with ERG on chromatin at the PSG genomic loci and PCa growth and progression. The β-Catenin signaling dependency can be pharmacologically targeted by small molecule inhibitors and LEF1/TCF O'PROTAC for the treatment of TMPRSS2-ERG/GOF mutant p53-positive PCa.

biological triplicates were sequenced by Illumina HiSeq 4000 following manufacture's protocol. Paired-end raw reads were subjected to the alignment of the mouse reference genome (GRCm38/mm10) using RNA-seq spliced read mapper STAR (v2.7.7a)[72]. Gene raw and normalized read counts were performed using RSeQC package (v2.3.6)[73]. Differential gene expression analysis was carried out by using DESeq2 (version 1.30.1)[74]. The false discovery rate (FDR) threshold 0.001 was applied to obtain the differentially genes.

### Chromatin immunoprecipitation (ChIP), ChIP-seq and ChIP-qPCR
DU145, LNCaP or VCaP cells were fixed with formaldehyde and subjected to sonication by Bioruptor (Diagenode) as described previously[71]. The supernatant was obtained and mixed with protein A/G beads and

antibodies for p53 (2 μg), ERG (2 μg) or β-Catenin (2 μg). After incubation overnight, beads were washed, and the complex containing DNA was incubated at 65 °C to reverse formaldehyde crosslinking. The eluted DNA was further treated with RNAase and proteinase K. Enriched DNA was extracted for high throughput sequencing or quantitative PCR.

For Re-ChIP assay, the immunoprecipitated DNA was processed twice for 15 min each time at 65 °C after the first-round ChIP by using elution buffer (50 mM TrispH 8.0, 1 mM EDTA, and 1% SDS). The eluted samples were diluted using the re-ChIP buffer (55 mM HEPES pH 7.9, 154 mM NaCl, 1.0 mM EDTA, 1.1% TritonX-100, 0.11% Na-deoxycholate) to make the final concentration of SDS lower to 0.1%. The second-round ChIP assay was performed as usual by adding the antibody into the immunoprecipitation mixture. Primers for ChIP-qPCR are listed in Supplementary Data 5.

For the ChIP-seq assay, sequencing libraries were prepared as previously described. The high-throughput sequencing was performed by Illumina HiSeq 4000 platform by the Mayo Clinic Genome Core Facilities. The raw reads were mapped to the human reference genome (GRCh37/hg38) using bowtie2 (version 2.2.9). MACS2 (version 2.1.1) was used for peak calling with a *p* value threshold of $1 \times 10^{-5}$. BigWig files were generated for visualization using the UCSC Genome Browser. The assignment of peaks to potential target genes was performed by the Genomic Regions Enrichment of Annotations Tool (GREAT). ERG ChIP-seq data generated from the mouse prostate tissues was downloaded from NCBI Gene Expression Omnibus (GEO) with accession number GSE47119[25]. β-Catenin ChIP-seq data was downloaded from GEO with accession number GSE53927[44], p53 ChIP-seq data from breast cancer cell lines were downloaded from GEO with accession number GSE59176[16].

### MEME-ChIP-seq DNA motif analysis of mutant p53 ChIP-seq peaks
We first extracted the 100-bp region centered around the summit of each peak called by MACS2[75]. We then extracted the DNA sequences of these 100-bp windows from the reference genome (GRCh38). Finally, we uploaded the DNA sequences to MEME-ChIP (https://meme-suite.org/meme/tools/meme-chip). MEME-ChIP reports up to 10 motifs if any.

### Chromosome conformation capture (3C) assay
The 3C assay was carried out according to the protocol as described[76]. Briefly, p53-KO DU145 cells transfected with indicated plasmids were crosslinked, collected and lysed. Chromation was digested with the indicated restriction enzymes. After digestion and ligation, DNA was purified and subjected to quantitative PCR (qPCR) analysis. *GAPDH* was used as an internal control. Primers are listed in Supplementary Data 5.

### Pyrimidine nucleotide measurement by liquid chromatography-mass spectrometry (LC-MS)
The analysis was performed by Mayo Metabolomic Core. Cell pellets were lysed in 1 × PBS by sonication. Proteins were removed by addition

of chilled acetonitrile/methanol. Supernatants were dried down and lipids were removed via Captiva EMR cartridges prior to LC-MS analysis. Using Agilent central carbon metabolites methods on an Agilent 6460 triple quadrupole mass spectrometer couple with a 1290 Infinity II quaternary pump, 20 analytes were captured in negative electrospray ionization and dynamic multiple reaction monitoring (dMRM) post ion-pairing reverse phase chromatographic separation. Uridine-5-monophosphate (323 > 79), deoxythymidine-5-diphosphate (401 > 79) and deoxythymidine-5-triphosphate (481 > 159) peaks were manually curated to confirm correct retention times. Peak areas of each analyte as well as protein content in each sample were reported.

### Generation and treatment of PCa xenografts in mice

Six-week SCID male mice were used in the study. Mice were housed in 22 °C, 55% humidity on average with a 12-h light/12-h dark cycle and access to food and water. Mouse experiments were approved by the Institutional Animal Care and Use Committee (IACUC) at the Mayo Clinic. The maximum of the xenograft is limited within 1 cm$^3$ according to the ethical requirement from IACUC. Mice were subcutaneously injected with VCaP cells ($5 \times 10^6$) mixed with Matrigel mixture (1 × PBS: Matrigel (BD Biosciences) = 1:1). After the xenografts reached a size of approximately 100 mm$^3$, mice were treated intraperitoneally with vehicle (90% corn oil (Sigma-Aldrich) + 10% DMSO), ICG-001 at 25 mg/kg for 5 days per week. For doxycycline (Dox)-inducible sh3 PSGs, mice were treated with 5% sucrose (-Dox) or 1 mg/ml doxycycline in 5% sucrose (+Dox) for indicated time points. For LEF1/TCF O'PROTAC administration, mice were transplanted with LuCaP23.1 PDX tumors in approximately the same volume. When the tumor volume reached approximately 100 mm$^3$, mice were randomly divided into three groups for treatment with PBS, control OP or LEF1 OP-V1 (10 mg/kg in PEI solution) via tail vein injection every other day. Mice were euthanized and tumor grafts were excised at the end of treatment. Tumor tissues were subjected to formalin fixation and paraffin embedding or lysed for protein or RNA extraction.

### Meta-analysis of ERG ChIP-seq generated from murine prostate tissues

ERG binding peaks were downloaded from the GEO (https://www.ncbi.nlm.nih.gov/geo/query/acc.cgi?acc=GSM1145303). These peaks were called by Chen et al. using MACS (v1.4) based on the mm9 reference genome[25]. We used GREAT's "basal + extension" rule to assign ERG peaks to the putative target genes[77]. Specifically, GREAT assigns each gene a basal regulatory element (i.e., from 5 Kb upstream to 1 Kb downstream of the transcription start site (TSS)) regardless of other nearby genes. A basal regulatory domain of a gene is extended in both directions to the nearest gene's basal domain but no more than 1000 Kb in one direction. The ERG ChIP-seq data was generated from the prostate tissues of $R26^{ERG}$ transgenic mice and the anti-ERG antibody from Epitomics (EPR3864) was used. Approximately 78% of ERG peaks are located in the distal regions (>5 Kb from the TSS).

### Meta-analysis of patient data

The status of *TP53* gene mutation/deletion or ERG fusion in PCa specimens from the cohorts of TCGA, SU2C and MSKCC was obtained through ciBioPortal (http://www.cbioportal.org/). The Z-score (FPKM) of *CTNNB1* and PSGs reflecting mRNA level in SU2C patient samples was downloaded and subjected to the comparison based on the status of *ERG* fusion or *TP53* gene alterations. Mann-Whitney *U* test was carried out to generate *p* value for the comparison.

### Statistics and Reproducibility

*P* values were determined by a χ2 test, two-tailed Student's *t* test, two-way ANOVA test, Fisher exact test or Mann-Whitney *U* test depending on the situations. All data are shown as mean values ± s.d. for experiments representing three or more independent experiments/replicates.

Cells and mice used in the study were randomly divided into the groups. Investigators collecting data and generating output were blinded to all groups. *P* values < 0.05 were considered statistically significant.

### Reporting summary

Further information on research design is available in the Nature Portfolio Reporting Summary linked to this article.

## Data availability

The RNA-seq and ChIP-seq data generated from the current study have been deposited in Gene Expression Omnibus (GEO) database with the accession number GSE184626. Source data are provided with this paper.

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

## Acknowledgements

The authors would like to thank Dr. Eva Cory from the University of Washington for kindly providing the LuCaP PDX models. This work was supported in part by the Mayo Clinic Foundation (to H.H.) and the National Institutes of Health (R01CA134514, R01CA130908 and R01CA271486 to H.H.).

## Author contributions

H.H. conceived the study. D.D., A.M.B., J.Z., Y.P., N.A.B. generated reagents and conducted experiment design and execution, data collection and data analysis. L.W. performed bioinformatics analysis of RNA-seq and ChIP-seq data. R.J. supervised mouse prostate histology and IHC analysis. H.H., D.D., L.J.M. wrote and edited the manuscript.

## Competing interests

The authors declare no competing interests.
