## [Peer Review File · Nature Communications]

REVIEWER COMMENTS

Reviewer #1 (Remarks to the Author); expert in prostate cancer and functional genomics:

This is an important study that challenges the existing paradigms on the role of mutant p53 in cancer. In this manuscript, Ding and colleagues showed the importance of GOF role of p53 in TMPRSS2-ERG gene fusion positive prostate cancer cells lines, LUCaP 23.1 PDX and GEMM models. Importantly, they demonstrated that mutations in the DNA binding domain of p53 promote new oncogenic transcriptional programs in T2-ERG positive prostate cancers. Using GEMM models of GOF mutant p53 (Pb-T2-ERG; TRP53R172H/-), they demonstrated that pyrimidine synthesis genes were specifically upregulated in the Pb-T2-ERG;TRP53R172H/- compared to Pb-T2-ERG;TRP53pc-/- . They found a new mutated p53 binding site that cooperatively binds with ERG in the CTNNB1 gene promoter and drives its expression. Further, they demonstrated that ERG/CTNNB1 axis regulates the expression of PSG's suggesting an indirect role for mP53 in regulating the PSG's. Finally, they employed small molecule inhibitors and oligonucleotide-based PROTACs to inhibit the b-catenin activity, suggesting an effective treatment of ERG/GOF mutant p53 positive prostate cancers. Overall, this manuscript was well written and interesting to read. Below are specific comments:

1. It is impressive to see such remarkable remodeling of the prostates in the Pb-T2-ERG; TRP53R172H/- mice. Were the CTNNB1 levels increased in this model?
2. As T2-ERG fusions and WNT pathway upregulation is observed in metastatic cancer, it is intriguing to check at the later time points (>15 months) for metastatic tumors? Further comment on the survival rate of the mice with adenocarcinomas.
3. It would be useful to determine whether the PSGs are upregulated even in the Pten/TRP53/ERG triple mutant model that the same group developed.
4. It is not clear whether the CTNNB1 gene expression is upregulated specifically by DBD p53 mutant cell lines? The authors should compare the expression levels of CTNNB1 and PSG's in a panel of WT, p53KO and DBD p53 mutant cell lines and perform correlation analysis with CTNNB1 and PSG's levels.
5. The authors demonstrated that ERG/mp53 drives the expression of CTNNB1. Please provide the exact position of the ERG/mp53 binding site with respect to the CTNNB1 TSS. It would be interesting to perform some luciferase assays using CTNNB1 gene promoter sequences and it is important to show the interaction of ERG/mp53 by PLA on the tissues/VCaP's that would support the findings.
6. In Fig 2b, it is intriguing that many of the binding effects of ERG are being mediated through its interaction with promoter elements. Which cell line was the ERG ChIP-seq generated from and which antibody was used? How many of the total binding sites of ERG map to distal regions?
7. In Fig 2c, 2g, the authors should include Pb-T2-ERG; Pb-T2-ERG;TRP53pc-/- group in their comparisons.
8. In Fig 2g/I, the authors demonstrate that only dual overexpression of ERG and mut-Trp53 lead to upregulation of Umps, Rrm1, etc. However, loss of ERG and p53 alone causes a dramatic reduction of these genes in VCaP. How do the authors explain this difference in the two instances?
9. The authors identify that the cistrome of the mutant p53 is ~ 400 sites. Is this true in the other datasets mentioned in the manuscript as well, ie: breast cancer dataset? Can the authors also plot RNA levels of Beta-catenin in other cancer datasets that harbor the p53 GoF mutant?
10. From Fig 4c, the expression levels of CTNNB1 among ERG fusion, TP53 mutations and ERG fusion+ TP53 mutations is not significant. Have the authors looked at the CTNNB1 level in the Pb-ERG/TP53 KO patients? It could serve as corroboratory evidence for the GEMM biology if there is a lack of increase in this subset. It would also be interesting to analyze the CTNNB1 levels based on the DBD p53 mutants and ERG fusion+ status in both SUC2 and MSKCC cohorts
11. Fig 8: It doesn't seem like p53 needs ERG to transactivate Beta-catenin. What do the authors believe is the role of ERG in the context of beta-catenin?

Reviewer #2 (Remarks to the Author); expert in p53:

In the manuscript " CTNNB1 1 transactivated by GOF mutant p53 instigates ERG pro-oncogene to drive pyrimidine synthesis and oncogenesis" Ding et al looked at the transactivation of CTNNB1 and examined if it relates to oncogenesis via ERG and pyrimidine synthesis. Using mouse model

systems, they show that TMPRSS2-ERG transgene together with Trp53 knockout induces prostate tumorigenesis in mice, which is accelerated by mutant Trp53 R172H knock-in. They also attempt to show GOF mutant p53 binds to a unique DNA sequence in CTNNB1 promoter and transactivates CTNNB1 gene expression. They attempt to show that β -Catenin is a driver and therapeutic target of ERG/GOF mutant p53-positive prostate cancer.

The manuscript claims one unique aspect of mutant p53's that has single amino acid changes in the DNA-binding domain (DBD) of p53- it can bind sequence specifically to a sequence of CTNNB1 promoter and this sequence does not bind to WT p53 or other p53 mutations outside the DBD. Only this aspect of the work is new/novel. Everything else in the manuscript has been described for various genes already, and thus do not deserve high impact. Thus, special attention needs to be provided to the DNA binding studies. Unless it is unequivocally proved that p53 mutants indeed binds sequence specifically and directly to a sequence different from WT p53 binding sequence and to which WT p53 does not bind, the manuscript does not shed any new light towards the biology of mutant p53 leading to oncogenesis. The major points that need to be addressed, as almost no concrete evidence exists in the literature to indicate that p53 mutants indeed can interact sequence specifically albeit with efforts in many laboratories (e.g., Moore et al. 1997 Int J Oncol. 1997 May;10(5):1035-45. doi: 10.3892/ijco.10.5.1035), are described below.

(1) If the authors are really trying to push the idea that p53-R248W is directly binding the sequence they identified, they need to perform a thorough TF analysis by ChIP etc to come to such a conclusion showing that no TF binding gets altered and is not involved. This is the case when they use a cell/nuclear extract.

(2) Direct interaction of mutant p53 needs to be shown by in vitro transcription and translation of the mutant p53 proteins and their interactions with the corresponding recognition site. Inclusion of a far-western analysis would strengthen the concept.

(3) Direct binding of mutant p53 with the recognized sequence would indicate that mutant p53 may not require any other factor for this binding. This needs to be demonstrated with purified untagged proteins that should then sequence specifically interact with the recognition site.

(4) Gel retention analysis must be done always with recognition site double-stranded sequences and mutated recognition site double-stranded sequences for competition assays. Use of different mutant sequences may establish the sequence specificity of mutant p53 and the recognition site.

(5) In the DNA-binding assays they must include a WT p53-binding site and WT p53 as a positive control.

(6) One important DNA binding assay that must be done in this context is the 'foot-print assay'. It can be done with purified protein from bacterial sources as the authors point out that bacterially expressed p53 mutants indeed bind the sequence specifically, although I am not sure if the assay included specific and non-specific competitor DNAs.

(7) The sequence "GCCCCCTCGCGCCCCGCCCCCTTGTC" was given in Figure 3 as the binding site for mutant p53. A cursory Blast of this sequence against human genome did not generate any exact match suggesting a more thorough examination of the identified sequences, although a specific search for the site near the CTNNB1 gene indeed finds the sequence. A thorough study of number such exact sites in the human genome by bioinformatic analysis in relation the possible genes they may regulate must be presented.

Other Significant Points:

(a) Lines 79-82. "Meta-analysis of the TCGA data showed that TMPRSS2-ERG fusion co-occurred with TP53 inactivation (mainly deletions) in primary/localized PCa patient samples (Fig. 1a,b), suggesting that both lesions are early events during PCa oncogenesis and may be involved in prostate tumorigenesis." --- It is not clear how the authors have come to the conclusion that both the lesions are early events.

(b) Lines 99-102. "To compare the effect of TP53 tumor suppressor gene deletion versus missense mutation on prostate oncogenesis, we also generated TMPRSS2-ERG transgenic and prostatic cell-specific Trp53 knockout/knockin compound mice (Pb-Cre+;Pb-T2-ERG;Trp53R172H /p (Pb-T2-ERG;Trp53pcR172H/-)) and their littermates (Trp53pcR172H/-) as control." --- It should have a p53-/- control.

(c) Lines 124-128. "This notion is further supported by the observation that TP53 gene mutation rate was about three-time higher in advanced PCa (e.g. ~36.7% in SU2C patients) compared to that in primary PCa (~12.5% in TCGA patients) (Extended Data Fig. 1c) and that both total and DBD mutations of TP53 co-occurred with ERG fusions in advanced PCa of the SU2C cohort (Fig.

1f)". -- Mutation rate: is it GOF p53 mutation rate or p53 gene mutation rate in general, including deletions? Also, do TCGA and other data support GOF p53 advantage?

(d) Lines 135-137. "Together, the data from both GEM models and cultured human PCa cell line indicate that missense p53 mutants such as R175H and R248W can act as GOF mutants to drive prostate oncogenesis."--- Very little data to substantiate this comment. No tumor data.

(e) Lines 171-174. "We identified the transcriptional coregulator binding pathway as the top enriched pathway (Fig. 3b) and detected a R248W-bound peak in the promoter of CTNNB1 gene which encodes β -Catenin, a core transcriptional component of the β -Catenin/TCF complexes (Fig. 3c)."---- What does p53-R172H (from mouse) data say?

(f) Lines 300-308. "Since PSGs are important for nucleotide 300 synthesis, we investigated whether expression of these PSGs play an important role in cell growth in TMPRSS2-ERG/ mutant p53 double positive PCa. Depletion of UMPS, RRM1 and RRM2, three key enzymes for pyrimidine synthesis (Fig. 2e) individually or together largely inhibited VCaP cell growth (Fig. 5i,j). To investigate the role of the PSGs in vivo, VCaP cells expressing doxycycline (Dox)-inducible PSG shRNAs were inoculated into SCID mice to generate xenograft tumors. We demonstrated that Dox-induced PSG knockdown largely inhibited VCaP tumor growth in mice (Fig. 5k,l). Dox administration also decreased tumor weight, but had no obvious effect on mouse body weight (Fig. 5m and Extended Data Fig. 7a)."--- What is the effect PSG knockdown in the growth of regular tissue or cells?

(g) Lines 335-337. "We demonstrated that ICG-001 administration significantly inhibited growth of VCaP xenograft tumors in mice but had little or no effect on mouse body weight (Fig. 6i-k and Extended Data Fig. 7e)". --- Do the tumors come back if you remove the drug?

(h) p53 mutation rate in PCa is relatively low as shown by many studies except in late stages when it reaches higher percentages only in metastatic cancer. Figure 1 states its percentage about 34% although later the authors admit that the percentage is 12% in TCGA; so, why the discrepancy?

(i) The authors do not differentiate between single amino acid substitutions and deletions at their will.

(j) The authors indicate that mutant p53 binding sites do not have any known transcription factor motif, but how large of sequence was used in defining the binding site? The peaks should be discrete and less than ~200 bp in size to keep the noise down when searching for statistically significant associations with known motifs. The peaks do not appear to be very discrete.

(k) The alteration of the binding sequence within the CTNNB1 promoter is a good idea, but the alteration needs to be precisely defined using a knock-in not knock-out. When deleting the sequence, it must be replaced with a dissimilar sequence of the same length so as not to alter the physical configuration of the promoter. The goal should be to alter the sequence such that mutant p53 binding could potentially be altered but not disrupt the normal CTNNB1 promoter function. The study in Figure 3 k and i doesn't address this issue; it looks like the deletions killed the promoter by potentially a multitude of ways.

Reviewer #3 (Remarks to the Author); expert in prostate cancer and beta-catenin:

In this manuscript the authors analyse patient prostate tumour data and argue that TMPRSS2-ERG gene fusions co-occur with TP53 mutations/deletions. To model this combination, they generate and analyse genetically modified mouse prostate tumours expressing the TMPRSS2-ERG rearrangement derived protein and TP53 deletion or mutation to establish whether this combination can induce prostate neoplasia. They show that substantial adenocarcinoma is observed when the mutant TP53 allele is combined with the TMPRSS2-ERG rearrangement derived protein. The phenotype is more advanced to that of combination mice with the TP53 deleted allele. Differential expression analysis of these tumours identified an upregulation of genes involved in pyrimide metabolism and the authors use various cell line models to investigate the molecular mechanisms that lead to this effect. They provide evidence to show that mutant p53 binds to sequences in the CTNNB1 locus and suggest that ERG and beta catenin regulate pyrimidine synthesis genes in prostate tumours with TMPRSS2-ERG gene fusions and TP53 mutations. They argue that beta-catenin is a therapeutic target for prostate cancers with these genetic alterations. Overall the quality of the data is very good and thorough and well presented. The association of ERG rearrangements and TP53 mutations as a prostate cancer driver is novel and the mechanistic

studies are convincing with complementary cellular models being used with a detailed molecular analysis. Whether beta catenin is a relevant therapeutic target in the clinic is less obvious but this is not the focus of the manuscript.

Major comments:

1. The human data analysis is somewhat superficial. The results from the study imply that TP53 deletions are acting differently to TP53 mutants to drive prostate cancer therefore comparisons and associations should be made with the different types of TP53 alterations separately. Additionally, studies have shown that TP53 mutants behave differently in a heterozygous versus a LOH state (Mc Cann et al *Oncogene*, 2022). This should be considered in the human sample analysis. What is the significance of including tumours with shallow deletions (Fig 1a) in the analysis? These presumably represent heterozygous wild type TP53, is the co-occurrence still significant when these are not considered?
2. In the prostate phenotypic analysis of the genetically modified mice, how was adenocarcinoma scored? The authors should include evidence of invasiveness using epithelial and smooth muscle markers in Figure 1C.
3. The authors compare the transcriptome from the double ERG and TP53 mutants versus the single genetic alterations. It would be good include the double ERG and TP53 deleted tumours in the validation qRTPCR assays (Figure 2C) to establish if the pathways analysed are divergent between the different TP53 alleles in the presence of ERG alterations.
4. The authors perform a p53 ChIPSeq in VCaP cells, which contain a mutant TP53, and show binding peaks within the CTNNB1 locus. As reports of direct DNA binding of mutant p53 have been varied, it would be good to confirm specificity by performing a ChIP assay on the shp53 cells shown in Fig 3m. The authors mention in the discussion that mutant TP53 has been shown to bind to promoters of nucleotide metabolism genes such as DCK, TK1 and IMPDH1, are these genes expressed at higher levels in the double mutant prostates and is there evidence of regulation of expression by mutant p53 in VCAP cells in experiments using shp53 cells?
5. Beta catenin has been implicated in WNT signalling, therefore some discussion and analysis of how this pathway fits with the model proposed should be done. What are the expression levels of WNT target genes such as AXIN2, LEF1 and MYC in VCaP and DU145 cells knocked out or down for ERG or mutant TP53? Immunohistochemistry for beta catenin should be done on prostates from double mutant animals to define localization within the prostate and within cells. Do patient tumours with mutant TP53 or dysregulated WNT signalling and ERG alterations express higher levels of pyrimide metabolism genes?
6. ERG has been shown to regulate WNT/LEF1 signalling at various levels in prostate cancer, with LEF1 identified as a direct target (Wu et al, *Cancer Res* 2013). This could mean that the involvement of ERG in the regulation of pyrimide metabolism genes could be at different levels and this point should be added to the discussion.
7. The authors show that inhibiting the expression of genes involved in pyrimide metabolism can affect growth in VCaP cells in vitro and in vivo. Can nucleotides rescue the growth defect of VCaP cells with shERG and shp53?
8. What other alterations are present in LuCaP 23.1 PDX that might impact the sensitivity in the O'PROTAC compound? As organoids are grown in factors involved in WNT signalling, the drug might not be specific to the TP53 and ERG alterations. This result would be more convincing if the treatment was also done on other organoid lines that do not carry TP53 and ERG alterations.

Minor point:

The concentrations of O'PROTAC compound in Figure 7 should be mentioned.

Reviewer #4 (Remarks to the Author); expert in nucleotide metabolism:

Remarks to the Author:

Ding and colleagues report the co-occurrence of TMPRSS2-ERG fusion gene with GOF p53 mutations in prostate cancer and associate this genotype to disease progression and aggressiveness. The authors show that ERG and GOF mtp53 cooperatively bind to the promoter of CTNNB1 gene, which in turn activates the expression of b-catenin. As a result, ERG and b-catenin promote the

expression of several key enzymes involved in pyrimidine biosynthesis, such as UMPS, RRM1, RRM2 and TYMS, which support the rapid demand for nucleotides of the cancer cells. Thus, pharmacological inhibition of b-catenin by small molecules as well as depletion/degradation by different means of downstream elements, such as LEFT1/TCF or PSG, importantly decrease tumor mass and volume, suggesting a therapeutic target in the treatment of TMPRSS2-ERG/GOF mtp53 prostate cancer. In this context, the manuscript nicely reports a very detailed mechanistic model which could potentially be exploited in the clinics. To this purpose, Ding et al. have included in vitro and in vivo models, as well as a comprehensive set of experiments to prove interaction to the promoter of the different genes and subsequent effects on protein expression and disease progression. As I'm not a specialist in the ChIP-seq analysis here reported, I won't discuss the experimental design, required controls or interpretation of these experiments. Taking that into account I found that overall, the manuscript is well written and contains novel information of relevance in the cancer field.

Major points:

- In Fig. 5g a reduction in dCtd and dUrd are observed upon depletion of ERG and p53, interpreted as a depletion in pyrimidine metabolites. Nevertheless, UMPS, RRM1/2 and TYMS are enzymes involved in de novo nucleotide biosynthesis, whereas dCtd and dUrd are either involved in the nucleotide salvage pathways or produced by catabolism of dCTP and dUTP, by the activities of SAMHD1 and dUTPase, respectively. As the aim of this experiment is to demonstrate that shERG and shp53 negatively affect PSG, I would rather analyse the dNTP levels (at least dCTP and dTTP), because they are the final products of this pathway.
- The authors show that inhibition of b-catenin by small molecules downregulates mRNA and protein expression of four PSGs in VCaP cells (Fig. 6c,f). However, as described in the manuscript, VCaP is TMPRSS2-ERG/GOF mtp53 double positive and all results are associated to this specific genotype, but no other cell line with a different genotype are shown in this experiment as a negative control. Thus, taking into account that nucleotide biosynthesis is usually upregulated in cancer cells, it could be very interesting to add other prostate cancer cell lines with a different genotype to corroborate whether b-catenin-dependent activation of PSGs are specific to the TMPRSS2-ERG/GOF mtp53 double positive cancer model or if, in contrast, b-catenin could regulate nucleotide levels in other prostate oncological diseases.
- In this manuscript, the authors demonstrate that the consequences of b-catenin inhibition and LEF1/TCF degradation on the tumoral progression are due to the depletion of PGS and thus of pyrimidine nucleotides (Fig 7). Several approved drugs exist targeting the activity of the enzymes encoded by the PGS, such as hydroxyurea and gemcitabine (against RRM), 5-fluorouracil (against TYMS) and pyrazofurin (against UMPS), so supporting the aforementioned hypothesis, I feel very curious to know the effects of those drugs in the development of the disease and whether they could have therapeutic applications in TMPRSS2-ERG/GOF mtp53 double positive prostate cancer.
- Last, regarding Fig. 7, though supplementation of dATP/dGTP and dTTP/dCTP revert the phenotype, I have some concerns. Deoxynucleoside triphosphates (dNTPs) are anionic molecules, which prevent passive diffusion over cellular membranes. For this reason, although many chemotherapies exploit the use of nucleotide analogues, they are administered as inactive nucleoside prodrugs, and reversion of dNTP-derived phenotypes are usually achieved through supplementation of nucleosides (in this case deoxyadenosine, deoxyguanosine, thymidine and deoxycytidine). Taking that into account, I didn't find the description of the nucleotide supplementation (which concentrations were used and the relative ratio of each nucleotide), and believe that the reversion of the phenotype is probably a result of the dNTP degradation to its nucleoside form prior entering the organoids.

Minor points:

- Line 150, "Upms" should be corrected to "Umps"
- Line 163, in "with 615 gene", correct to "genes"
- In Figure 3m, I guess that the third column in the western blot should have a "+" in the shCon row instead of a "-"

RESPONSE TO REVIEWERS' COMMENTS

We thank the Reviewers for the time they spent in evaluating our work and for their insightful comments, which we have considered thoroughly in generating the revised manuscript.

REVIEWER COMMENTS

Reviewer #1 (Remarks to the Author); expert in prostate cancer and functional genomics:

This is an important study that challenges the existing paradigms on the role of mutant p53 in cancer. In this manuscript, Ding and colleagues showed the importance of GOF role of p53 in TMPRSS2-ERG gene fusion positive prostate cancer cells lines, LUCaP 23.1 PDX and GEMM models. Importantly, they demonstrated that mutations in the DNA binding domain of p53 promote new oncogenic transcriptional programs in T2-ERG positive prostate cancers. Using GEMM models of GOF mutant p53 (Pb-T2-ERG; TRP53R172H/-), they demonstrated that pyrimidine synthesis genes were specifically upregulated in the Pb-T2-ERG;TRP53R172H/- compared to Pb-T2-ERG;TRP53pc-/-. They found a new mutated p53 binding site that cooperatively binds with ERG in the CTNNB1 gene promoter and drives its expression. Further, they demonstrated that ERG/CTNNB1 axis regulates the expression of PSG's suggesting an indirect role for mP53 in regulating the PSG's. Finally, they employed small molecule inhibitors and oligonucleotide-based PROTACs to inhibit the b-catenin activity, suggesting an effective treatment of ERG/GOF mutant p53 positive prostate cancers. Overall, this manuscript was well written and interesting to read. Below are specific comments:

Reply: We thank the Reviewer for the positivity and recognizing the significance of our finding. We also thank the Reviewer for the insightful suggestions that have helped us improve our manuscript significantly.

1. It is impressive to see such remarkable remodeling of the prostates in the Pb-T2-ERG; TRP53R172H/- mice. Were the CTNNB1 levels increased in this model?

Reply: We appreciate the insightful comments. We examined mouse β -Catenin mRNA expression in six groups of mice as shown in Fig. 1c. We found that Ctnnb1 mRNA expression was significantly increased in prostate tumors of Pb-T2-ERG;Trp53R172H/- mice compared to other five groups of mice (Fig. 4c). IHC analysis showed that β -Catenin protein was also substantially increased in prostate tumors from Pb-T2-ERG;Trp53R172H/- mice (Supplementary Fig. 2d, Bottom). These results indicate that β -Catenin expression was upregulated in prostate tumors of Pb-T2-ERG;Trp53R172H/- mice.

2. As T2-ERG fusions and WNT pathway upregulation is observed in metastatic cancer, it is intriguing to check at the later time points (>15 months) for metastatic tumors? Further comment on the survival rate of the mice with adenocarcinomas.

Reply: We did not find any visible local metastasis and distant metastasis in different organ sites such as lymph node, liver and lung in T2-ERG/Trp53R172H/- mice at the later time points (>15 months) when large tumors developed in the prostate. As suggested by the Reviewer, we

provided mouse survival rate data in the revised manuscript. As shown in Fig. 1f, the survival rate of Pb-T2-ERG; Trp53R172H/- mice was much worse compared to other five groups. Despite that we did not observe obvious tumor metastasis in this mouse genotype, we did often see bladder obstructions which could be a plausible explanation for the poor survival rate of these mice compared to other genotypes.

3. It would be useful to determine whether the PSGs are upregulated even in the Pten/TRP53/ERG triple mutant model that the same group developed.

Reply: As suggested, we examined the expression of PSGs in the Pten/Trp53^{R172H/-} (TRP53)/ERG triple mutant model we developed previously (Blee et al., Clin Cancer Res 24: 4551-4565, 2018). We found that the expression of PSGs was higher in Pten/TRP53/ERG triple mutant model compared to WT mice (Supplementary Fig. 4a-e), consistent with a role of GOF mutant p53 and ERG in regulation of expression of PSGs.

4. It is not clear whether the CTNNB1 gene expression is upregulated specifically by DBD p53 mutant cell lines? The authors should compare the expression levels of CTNNB1 and PSG's in a panel of WT, p53KO and DBD p53 mutant cell lines and perform correlation analysis with CTNNB1 and PSG's levels.

Reply: We examined CTNNB1 expression in VCaP (T2-ERG; p53 R248W), C4-2 (p53 WT), LNCaP (p53 WT), PC-3 (p53 null), 22RV1 (p53 Q331R, a mutant in TET domain), DU145 (p53 R223L/V274F). The new data showed that CTNNB1 expression level was much higher in DBD p53 mutant cell lines VCaP and DU145 compared to other cell lines (Supplementary Fig. 6f). As we reported in this study, the upregulation of PSGs' expression not only requires p53 mutant but also ERG overexpression. Accordingly, we found that compared to other cell lines examined expression of PSGs was much higher in VCaP cells which harbor p53 DBD mutation R248W and TMPRSS2-ERG fusion (ERG overexpression) (Supplementary Fig. 8a).

5. The authors demonstrated that ERG/mp53 drives the expression of CTNNB1. Please provide the exact position of the ERG/mp53 binding site with respect to the CTNNB1 TSS. It would be interesting to perform some luciferase assays using CTNNB1 gene promoter sequences and it is important to show the interaction of ERG/mp53 by PLA on the tissues/VCaP's that would support the findings.

Reply: As shown in Fig. 3e, the exact position of mp53 binding site in the CTNNB1 gene promoter is from -63 to -39 relative to the transcriptional start site (TSS, designated as position +1). As shown in Fig. 4e, the exact position of ERG binding site in the CTNNB1 gene promoter is from -28 to -25 relative to the TSS.

As suggested by the Reviewer, we performed the luciferase assays using CTNNB1 gene promoter sequences in p53-null PC-3 cells transfected with different p53 mutants individually in combination with ERG. Our new data showed that expression of p53 DBD mutants R175H, C238Y, R248W or R273H, but the TET domain mutant Q331R largely increased the luciferase activity of reporter gene and importantly, the luciferase activity was largely increased after co-

transfection of ERG with each p53 DBD mutant (Fig. 4j). We also demonstrate the interaction between ERG and mutant p53 in VCaP cells by PLA assay (Supplementary Fig. 5d).

6. In Fig 2b, it is intriguing that many of the binding effects of ERG are being mediated through its interaction with promoter elements. Which cell line was the ERG ChIP-seq generated from and which antibody was used? How many of the total binding sites of ERG map to distal regions?

Reply: We downloaded the ERG binding peaks from the GEO (<https://www.ncbi.nlm.nih.gov/geo/query/acc.cgi?acc=GSM1145303>). These peaks were called by Chen et al. using MACS (v1.4) based on the mm9 reference genome (Chen et al., Nat Med, 19: 1023-9, 2013). We used GREAT's "basal + extension" rule to assign ERG peaks to the putative target genes (McLean et al., Nat Biotechnol, 28: 495-501, 2010). Specifically, GREAT assigns each gene a basal regulatory element (i.e., from 5Kb upstream to 1Kb downstream of the TSS) regardless of other nearby genes. Then, a gene's basal regulatory domain is extended in both directions to the nearest gene's basal domain but no more than 1000 Kb in one direction. As a result, 24665 peaks were assigned to 13063 genes, which are the number of ERG bound genes shown in Fig. 2b.

The ERG ChIP-seq data was generated from the prostate tissues of *R26^{ERG}* transgenic mice and the anti-ERG antibody from Epitomics (EPR3864) was used. Approximately 78% of ERG peaks are located in distal regions (> 5Kb from the TSS). This new information has been added in a new section "Meta-analysis of ERG ChIP-seq generated from murine prostate tissues" in Methods in Supplementary Information.

7. In Fig 2c, 2g, the authors should include Pb-T2-ERG; Pb-T2-ERG;TRP53pc^{-/-} group in their comparisons.

Reply: We thank the Reviewer for the suggestion. We compared expression of PSGs in Trp53pc^{-/-} and Pb-T2-ERG;Trp53pc^{-/-} mice to their expression in other four groups of mice. Our new data showed that the expression levels of PSGs were much higher in prostate tumors of Pb-T2-ERG;Trp53pcRH^{-/-} mice compared to prostate tissues from Pb-T2-ERG;Trp53pc^{-/-} mice and other genotype mice (Fig. 2c,g,h), further supporting a role of GOF mutant of p53 in regulating expression of PSGs in the prostate.

8. In Fig 2g/I, the authors demonstrate that only dual overexpression of ERG and mut-Trp53 lead to upregulation of Umps, Rrm1, etc. However, loss of ERG and p53 alone causes a dramatic reduction of these genes in VCaP. How do the authors explain this difference in the two instances?

Reply: That is a great point. As indicated in the manuscript, knockdown of either ERG or p53 mutant could inhibit the expression of PSGs and this is because both ERG and p53 mutant are necessarily required for PSG overexpression and therefore loss of each of them jeopardized the proper expression of PSGs in VCaP cells.

9. The authors identify that the cistrome of the mutant p53 is ~ 400 sites. Is this true in the other datasets mentioned in the manuscript as well, ie: breast cancer dataset? Can the authors also plot RNA levels of Beta-catenin in other cancer datasets that harbor the p53 GoF mutant?

Reply: We thank the Reviewer for the excellent points. As shown in Fig. 3a in the previous submission, we identified 1116 common mutant p53 binding sites from two replicates in VCaP cells. As reported previously, there were 2177 WT p53 binding sites in MCF-7 cells (GSM1429753), 2320 WT p53 binding sites in MDA-MB-175-VII cells (GSM1429754), 2706 p53 R273H binding sites in MDA-MB-468 cells (GSM1429755), 1694 p53 R248Q binding sites in HCC70 cells (GSM1429756), and 1500 p53 R249S binding sites in BT-549 cells (GSM1429757). Therefore, in terms of the total binding sites, the p53 mutant R248W binding sites we identified in VCaP cells were lower compared to these studies reported previously, but was also in the 1-2k range and the difference could be due to the cell line difference.

As suggested, we also examined expression of CTNNB1 in other cancer datasets that harbor the p53 GoF mutant, including bladder cancer, colon cancer, breast cancer and pancreatic cancer. We found that CTNNB1 expression was not significantly upregulated in p53 mutated bladder and colon cancer patient samples (Supplementary Fig. 6i,j); however, significant upregulation of CTNNB1 was observed in p53 mutant breast and pancreatic cancer patient samples (Supplementary Fig. 6k,l). The data from breast and pancreatic cancers provide further support to our finding in prostate cancer that GoF p53 mutant is a critical factor for CTNNB1 expression; however, the data from bladder and colon cancers suggest that mutant p53 regulation of CTNNB1 may also be influenced by other factors such as ERG in prostate cancer cells.

10. From Fig 4c, the expression levels of CTNNB1 among ERG fusion, TP53 mutations and ERG fusion+ TP53 mutations is not significant. Have the authors looked at the CTNNB1 level in the Pb-ERG/TP53 KO patients? It could serve as corroboratory evidence for the GEMM biology if there is a lack of increase in this subset. It would also be interesting to analyze the CTNNB1 levels based on the DBD p53 mutants and ERG fusion+ status in both SUC2 and MSKCC cohorts.

Reply: Fig. 4c (now is Fig. 4d in the revised manuscript) showed CTNNB1 expression level was higher in the ERG fusion+ TP53 mutation prostate cancer patient samples compared to p53 WT patient samples. Importantly, the expression of CTNNB1 was significantly higher in ERG fusion and TP53 mutation patient samples than that in ERG fusion positive, but TP53 null patient samples, further highlighting the importance of GoF mutant of p53 in regulating CTNNB1 expression. Indeed, these data were derived from the SU2C cohort since tumors in this cohort (with advanced/metastatic prostate cancer) tend to have more p53 mutations than tumors in the TCGA cohort. We did not evaluate the level of CTNNB1 in MSKCC cohort since this cohort does not have RNA-seq data.

11. Fig 8: It doesn't seem like p53 needs ERG to transactivate Beta-catenin. What do the authors believe is the role of ERG in the context of beta-catenin?

Reply: We thank the Reviewer for the great point. We utilized the scheme shown in Fig. 8 to show the point that p53 plays an important role in transcription of *CTNNB1* while ERG-binding

on the promoter of *CTNNB1* also contributes significantly to the expression of *CTNNB1* gene. Such role of ERG as indicated in the model is also supported by the impact of ERG on H3K27ac level in the *CTNNB1* gene promoter in VCaP cells (Fig. 4i).

Reviewer #2 (Remarks to the Author); expert in p53:

In the manuscript "CTNNB1 1 transactivated by GOF mutant p53 instigates ERG pro-oncogene to drive pyrimidine synthesis and oncogenesis" Ding et al looked at the transactivation of CTNNB1 and examined if it relates to oncogenesis via ERG and pyrimidine synthesis. Using mouse model systems, they show that TMPRSS2-ERG transgene together with Trp53 knockout induces prostate tumorigenesis in mice, which is accelerated by mutant Trp53 R172H knock-in. They also attempt to show GOF mutant p53 binds to a unique DNA sequence in CTNNB1 promoter and transactivates CTNNB1 gene expression. They attempt to show that β -Catenin is a driver and therapeutic target of ERG/GOF mutant p53-positive prostate cancer. The manuscript claims one unique aspect of mutant p53's that has single amino acid changes in the DNA-binding domain (DBD) of p53- it can bind sequence specifically to a sequence of CTNNB1 promoter and this sequence does not bind to WT p53 or other p53 mutations outside the DBD. Only this aspect of the work is new/novel. Everything else in the manuscript has been described for various genes already, and thus do not deserve high impact. Thus, special attention needs to be provided to the DNA binding studies. Unless it is unequivocally proved that p53 mutants indeed binds sequence specifically and directly to a sequence different from WT p53 binding sequence and to which WT p53 does not bind, the manuscript does not shed any new light towards the biology of mutant p53 leading to oncogenesis. The major points that need to be addressed, as almost no concrete evidence exists in the literature to indicate that p53 mutants indeed can interact sequence specifically albeit with efforts in many laboratories (e.g., Moore et al. 1997 Int J Oncol. 1997 May;10(5):1035-45. doi: 10.3892/ijo.10.5.1035), are described below.

Reply: We thank the Reviewer for the positivity and recognizing novel aspects of our findings. We also thank the Reviewer for the insightful suggestions that have helped us improve our manuscript significantly.

(1) If the authors are really trying to push the idea that p53-R248W is directly binding the sequence they identified, they need to perform a thorough TF analysis by ChIP etc to come to such a conclusion showing that no TF binding gets altered and is not involved. This is the case when they use a cell/nuclear extract.

Reply: We thank the Reviewer for the suggestion. In Supplementary Fig. 5f, the search for TF motif in the mutant p53 binding sites was performed. Even though two consensus motifs were identified, but they did not match to any motifs bound by known TFs. We also examined the binding sequence of mutant p53 using the P-Match algorithm (<http://gene-regulation.com/pub/programs.html#pmatch>). As a result, again we did not identify any transcriptional factors that could recognize DNA sequences in mutant p53 binding sites.

(2) Direct interaction of mutant p53 needs to be shown by in vitro transcription and translation of the mutant p53 proteins and their interactions with the corresponding recognition site. Inclusion of a far-western analysis would strengthen the concept.

Reply: Our reciprocal co-IP assays showed that endogenous ERG and mutant p53 (R248W) proteins were present in the same complex in VCaP cells; however, their interaction was abolished by adding ethidium bromide (EtBr) (Supplementary Fig. 5c). These data suggest that ERG-mutant p53 interaction could be mediated by DNA. Indeed, their interaction in the nucleus was further confirmed by PLA assay as required by Reviewer #1 (Supplementary Fig. 5d).

As suggested by the Reviewer, we examined the interaction between ERG and mutant p53 by performing protein binding assay using in vitro translated ERG and FLAG-tagged p53 mutants or FOXO1 (positive control as reported by Yang et al., *Cancer Res*, 77: 6524-6537, 2017). As shown in Supplementary Fig. 5e, while ERG bond to FOXO1, it didn't bind to any of p53 mutants examined. These data provide further support to the notion that ERG interacts with mutant p53 indirectly.

(3) Direct binding of mutant p53 with the recognized sequence would indicate that mutant p53 may not require any other factor for this binding. This needs to be demonstrated with purified untagged proteins that should then sequence specifically interact with the recognition site.

Reply: Thank you for the suggestion. We first purified GST-tagged wild-type or mutant p53 and then cleaved GST off using thrombin. Thrombin was removed through incubation of protein solution with Benzamidine Sepharose 6B. The purified untagged wild-type and mutant p53 was subjected to the EMSA assay. As shown in Figure 3j, all DBD mutants, but not TET mutant or WT p53 bound MP53BS from the *CTNNB1* gene promoter and the binding was completely blocked by unlabeled probe.

(4) Gel retention analysis must be done always with recognition site double-stranded sequences and mutated recognition site double-stranded sequences for competition assays. Use of different mutant sequences may establish the sequence specificity of mutant p53 and the recognition site.

Reply: Excellent point. As suggested, we generated mutated probe by mutating the first and second cytosine-rich motif individually or together (Supplementary Fig. 6d). New EMSA assays showed that only mutations in both cytosine-rich motifs abolished mutant p53 binding to the MP53BS in the *CTNNB1* gene promoter (Supplementary Fig. 6e).

(5) In the DNA-binding assays they must include a WT p53-binding site and WT p53 as a positive control.

Reply: We performed new EMSA assay using nuclear extract (NE) of VCaP (p53 R248W GoF mutant) and LNCaP (WT p53) and MP53BS and WT p53 binding sequence in CDKN1A as probes. Our new data showed that MP53BS probe was only bound by NE from VCaP, but not LNCaP and that CDKN1A probe was only bound by NE from LNCaP, but not VCaP (Supplementary Fig. 5h), indicating that MP53BS is only specifically bound by mutant p53, but not WT p53.

(6) One important DNA binding assay that must be done in this context is the 'foot-print assay'. It can be done with purified protein from bacterial sources as the authors point out that

bacterially expressed p53 mutants indeed bind the sequence specifically, although I am not sure if the assay included specific and non-specific competitor DNAs.

Reply: We agree with the Reviewer that the ‘foot-print assay’ is one important DNA binding assay to perform. The use of radioisotope-labeled DNA probe is essential for successful foot-print assay. Our lab has not used radioisotope since 2011 and we need to have the radioisotope use permit first in order for us to perform the experiments. We have submitted the application; however, the step of background check of radioisotope users takes much longer time than we thought and therefore, so far we have not had our radioisotope use permit granted yet. While we have not been able to perform this critical experiment due to this administration issue, we instead performed a series of other essential studies to support our conclusion as outlined below:

- 1) Our ChIP-seq data from prostate cancer cell line VCaP (p53 R248W mutant) and others done in breast cancer cell lines (with p53 mutants and WT p53) (Zhu, J. *et al. Nature* **525**, 206-211 (2015) showed that only mutated p53, but not WT p53 bound to the CTNNB1 gene promoter (Fig. 3c and Supplementary Fig. 5b). , it was clearly suggested that p53 mutant could bind to the indicated sequence (MP53BS).
- 2) We performed EMSA using MP53BS as probe and nuclear extract of VCaP cells and an obvious DNA/protein complex was detected and importantly, this complex was disrupted by adding extra amount of unlabeled MP53BS probe in the EMSA reaction (Fig. 3h), suggesting the specific binding of MP53BS.
- 3) We performing the supershift assay, we demonstrated that MP53BS probe binding signals in VCAP nuclear extract was competed away (supershift) by a p53 mutant-recognizing antibody (Fig. 3i).
- 4) We purified GST-p53 WT, DBD mutants and a TET mutant from bacteria and GST tag was removed and these purified proteins were used for new EMSA. We found that all the p53 DBD mutants examined, but not the TET mutant or WT p53 bound to MP53BS and importantly, the binding signal was completely competed away after adding the unlabeled MP53BS probe (Fig. 3j) suggesting a specific direct binding of p53 DBD mutants with MP53BS.
- 5) We generated several MP53BS deletion clones from p53 DBD mutant (R223L/V274F) cell line DU145 using CRISPR/Cas9 (Supplementary Fig. 6c). We found that deletion of this MP53BS motif in the CTNNB1 gene promoter largely diminished CTNNB1 mRNA and β -Catenin protein expression in DU145 cells (Fig. 3k,l) and importantly, we found that co-knockdown of mutant p53 in DU145 cells did not result in further reduction in CTNNB1 mRNA and β -Catenin protein expression (Fig. 3m,n).
- 6) In addition to the deletion approach, we also mutated two cytosine-rich motifs into A/T nucleotides in MP53BS and performed new EMSA. We found that only mutations in both cytosine-rich motifs abolished MP53BS binding in the VCaP nuclear extract (Supplementary Fig. 6d,e).

Together, our existent and new data from a series of experiments consistently suggest that MP53BS can be specifically recognized by the p53 DBD mutant(s), but not WT p53. Because we have not been able to perform the critical ‘foot-print assay’ as requested by the Reviewer, we discussed this situation with Dr. Maria Garcia Fernandez and she indicated that it is appropriate for us to emphasize that future DNase I footprinting studies are important for further assessment of the binding of MP53BS by p53 mutants and acknowledge this as a limitation of the current

study. We have acknowledged this limitation in the revised manuscript on page 26.

(7) The sequence “GCCCCCTCGCGCCCCGCCCCTTGTC” was given in Figure 3 as the binding site for mutant p53. A cursory Blast of this sequence against human genome did not generate any exact match suggesting a more thorough examination of the identified sequences, although a specific search for the site near the CTNNB1 gene indeed finds the sequence. A thorough study of number such exact sites in the human genome by bioinformatic analysis in relation the possible genes they may regulate must be presented.

Reply: As suggested, we performed bioinformatic analysis of *CTNNB1* MP53BS-like sequences in other mutant p53 binding targets. We found that 288 mutant p53 binding target genes contain *CTNNB1* MP53BS-like sequences and the data are provided in Supplementary Table 4.

Other Significant Points:

(a) Lines 79-82. “Meta-analysis of the TCGA data showed that TMPRSS2-ERG fusion co-occurred with TP53 inactivation (mainly deletions) in primary/localized PCa patient samples (Fig. 1a,b), suggesting that both lesions are early events during PCa oncogenesis and may be involved in prostate tumorigenesis.” --- It is not clear how the authors have come to the conclusion that both the lesions are early events.

Reply: We agree with the Reviewer and we have omitted the point of “early events” in our statement.

(b) Lines 99-102. “To compare the effect of TP53 tumor suppressor gene deletion versus missense mutation on prostate oncogenesis, we also generated TMPRSS2-ERG transgenic and prostatic cell-specific Trp53 knockout/knockin compound mice (Pb-Cre+;Pb-T2-ERG;Trp53R172H /p (Pb-T2-ERG;Trp53pcR172H/-)) and their littermates (Trp53pcR172H/-) as control.” --- It should have a p53^{-/-} control.

Reply: In the previous version of the manuscript, we described ERG/Trp53^{-/-} (KO) mice and ERG/Trp53R172H^{-/-} (KI) mice in two separate paragraphs. To make direct comparisons between these two groups of mice, we now described all six groups of GEM mice including ERG/Trp53 KO and ERG/Trp53 KI mice at the same time on page 5 in the revised manuscript (Fig. 1c-e and Supplementary Fig. 2a-d).

(c) Lines 124-128. “This notion is further supported by the observation that TP53 gene mutation rate was about three-time higher in advanced PCa (e.g. ~36.7% in SU2C patients) compared to that in primary PCa (~12.5% in TCGA patients) (Extended Data Fig. 1c) and that both total and DBD mutations of TP53 co-occurred with ERG fusions in advanced PCa of the SU2C cohort (Fig. 1f)”. -- Mutation rate: is it GOF p53 mutation rate or p53 gene mutation rate in general, including deletions? Also, do TCGA and other data support GOF p53 advantage?

Reply: We are sorry for not being clear. The rate of TP53 mutations (~12.5% in TCGA patients and ~36.7% in SU2C patients) indicated in Supplementary Fig. 1c (Supplementary Fig. 1e in the revised manuscript) was not the GOF p53 mutation rate, but the rate of TP53 gene single amino

acid substitution mutations including missense, truncating and splice mutations. We have included this information in the main text.

(d) Lines 135-137. “Together, the data from both GEM models and cultured human PCa cell line indicate that missense p53 mutants such as R175H and R248W can act as GOF mutants to drive prostate oncogenesis.”--- Very little data to substantiate this comment. No tumor data.

Reply: We agree and have modified the statement based on the data we presented.

(e) Lines 171-174. “We identified the transcriptional coregulator binding pathway as the top enriched pathway (Fig. 3b) and detected a R248W-bound peak in the promoter of CTNNB1 gene which encodes β -Catenin, a core transcriptional component of the β -Catenin/TCF complexes (Fig. 3c).”---- What does p53-R172H (from mouse) data say?

Reply: To address this concern, we generated organoids from murine prostate tumors of T2-ERG/Trp53R172H/- mice. Our ChIP-qPCR analysis showed that Trp53 R172H mutant bound to the murine Ctnnb1 gene promoter in T2-ERG/Trp53R172H/- mouse prostate cancer cells (Supplementary Fig. 5k).

(f) Lines 300-308. “Since PSGs are important for nucleotide 300 synthesis, we investigated whether expression of these PSGs play an important role in cell growth in TMPRSS2-ERG/ mutant p53 double positive PCa. Depletion of UMPS, RRM1 and RRM2, three key enzymes for pyrimidine synthesis (Fig. 2e) individually or together largely inhibited VCaP cell growth (Fig. 5i,j). To investigate the role of the PSGs in vivo, VCaP cells expressing doxycycline (Dox)-inducible PSG shRNAs were inoculated into SCID mice to generate xenograft tumors. We demonstrated that Dox-induced PSG knockdown largely inhibited VCaP tumor growth in mice (Fig. 5k,l). Dox administration also decreased tumor weight, but had no obvious effect on mouse body weight (Fig. 5m and Extended Data Fig. 7a).”--- What is the effect PSG knockdown in the growth of regular tissue or cells?

Reply: We knocked out three PSGs including UMPS, RRM1 and RRM2 individually in two benign/’normal’ prostatic cell lines BPH1 and RWPE1. We found that depletion of each gene alone did modestly reduce the growth of these cell lines (Supplementary Fig. 8b-e). However, it is worth noting that the growth inhibitory scope caused by knockout of these genes in benign prostatic cells was much smaller than that seen in ERG fusion/TP53 mutant prostate cancer cells (Fig. 5i-m). These data suggest that ERG fusion/TP53 mutant prostate cancer cells rely more on the expression of PSGs than the regular/benign prostate cells.

(g) Lines 335-337. “We demonstrated that ICG-001 administration significantly inhibited growth of VCaP xenograft tumors in mice but had little or no effect on mouse body weight (Fig. 6i-k and Extended Data Fig. 7e)”. --- Do the tumors come back if you remove the drug?

Reply: We repeated the in vivo experiment by including a group in which ICG-001 was removed at 11 days after drug treatment. Our new data showed that the tumors largely came back after drug removal (Supplementary Fig. 9l,m). These results are consistent with the observations that ICG-001 inhibited cell proliferation in vitro and Ki67 expression in tumors in mice (Fig. 6e, l, m).

(h) p53 mutation rate in PCa is relatively low as shown by many studies except in late stages when it reaches higher percentages only in metastatic cancer. Figure 1 states its percentage about 34% although later the authors admit that the percentage is 12% in TCGA; so, why the discrepancy?

Reply: To avoid the obvious discrepancy, in the main text of the revised manuscript we explicitly indicated that 35% of prostate tumors in TCGA had TP53 gene mutation and/or homozygous or heterozygous deletions (Fig. 1a) and that 12.5% of TCGA tumors had gene mutations including missense, truncation and splice mutations (Supplementary Fig. 1e, Top).

(i) The authors do not differentiate between single amino acid substitutions and deletions at their will.

Reply: As indicated above, we have explicitly distinguished single amino acid substitutions from deletions in the revised manuscript.

(j) The authors indicate that mutant p53 binding sites do not have any known transcription factor motif, but how large of sequence was used in defining the binding site? The peaks should be discrete and less than ~200 bp in size to keep the noise down when searching for statistically significant associations with known motifs. The peaks do not appear to be very discrete.

Reply: We agree with the Reviewer that smaller and discrete peaks increase the signal-to-noise ratio during motif search. In our study, the median size of peaks identified from mutant p53 ChIP-seq is approximately 250 bp (median size = 244 and 253 bp in each of two replicates, respectively). Indeed, we took the 100 bp window centered around the summit of each peak to conduct the motif search. A detailed section termed “MEME-ChIP-seq DNA motif analysis of mutant p53 ChIP-seq peaks” has been added to the Methods section in the Supplementary Information.

(k) The alteration of the binding sequence within the CTNNB1 promoter is a good idea, but the alteration needs to be precisely defined using a knock-in not knock-out. When deleting the sequence, it must be replaced with a dissimilar sequence of the same length so as not to alter the physical configuration of the promoter. The goal should be to alter the sequence such that mutant p53 binding could potentially be altered but not disrupt the normal CTNNB1 promoter function. The study in Figure 3 k and i doesn't address this issue; it looks like the deletions killed the promoter by potentially a multitude of ways.

Reply: These are excellent points. We performed additional EMSA assays by comparing the binding capability of unmutated MP53BS to three mutants with dissimilar sequences but the same length. Consistent with the finding in cells that deletion of MP53BS largely diminished expression of β -Catenin at both protein and mRNA levels (Fig. 3k,l), we found that mutations of two cytosine-rich motifs in MP53BS abolished the binding ability of MP53BS (Supplementary Fig. 6d,e). Thus, similar to the effect of deletion mutant, altering the p53 binding sequence also altered mutant p53 binding, indicating that the disrupted effects of MP53BS mutations were less likely due to the potential alterations in physical configuration.

Reviewer #3 (Remarks to the Author); expert in prostate cancer and beta-catenin:

In this manuscript the authors analyse patient prostate tumour data and argue that TMPRSS2-ERG gene fusions co-occur with TP53 mutations/deletions. To model this combination, they generate and analyse genetically modified mouse prostate tumours expressing the TMPRSS2-ERG rearrangement derived protein and TP53 deletion or mutation to establish whether this combination can induce prostate neoplasia. They show that substantial adenocarcinoma is observed when the mutant TP53 allele is combined with the TMPRSS2-ERG rearrangement derived protein. The phenotype is more advanced to that of combination mice with the TP53 deleted allele. Differential expression analysis of these tumours identified an upregulation of genes involved in pyrimide metabolism and the authors use various cell line models to investigate the molecular mechanisms that lead to this effect. They provide evidence to show that mutant p53 binds to sequences in the CTNNB1 locus and suggest that ERG and beta catenin regulate pyrimidine synthesis genes in prostate tumours with TMPRSS2-ERG gene fusions and TP53 mutations. They argue that beta-catenin is a therapeutic target for prostate cancers with these genetic alterations. Overall the quality of the data is very good and thorough and well presented. The association of ERG rearrangements and TP53 mutations as a prostate cancer driver is novel and the mechanistic studies are convincing with complementary cellular models being used with a detailed molecular analysis. Whether beta catenin is a relevant therapeutic target in the clinic is less obvious but this is not the focus of the manuscript.

Reply: We thank the Reviewer for the positivity and recognizing the quality of our data and significance of our findings. We also thank the Reviewer for the insightful suggestions that have helped us improve our manuscript significantly.

Major comments:

1. The human data analysis is somewhat superficial. The results from the study imply that TP53 deletions are acting differently to TP53 mutants to drive prostate cancer therefore comparisons and associations should be made with the different types of TP53 alterations separately. Additionally, studies have shown that TP53 mutants behave differently in a heterozygous versus a LOH state (Mc Cann et al Oncogene, 2022). This should be considered in the human sample analysis. What is the significance of including tumours with shallow deletions (Fig 1a) in the analysis? These presumably represent heterozygous wild type TP53, is the co-occurrence still significant when these are not considered?

Reply: We performed new analyses by stratifying tumors into different groups based on the status of TP53 gene deletions and/or mutations including the heterozygous (or shallow) deletion (TP53^{WT/-}). We found that ERG fusion and TP53 complete loss (TP53^{-/-}), but not TP53 shallow deletion (TP53^{WT/-}) co-occurred in TCGA data although the co-occurrence was just shy of statistically significance (P = 0.069), and this could be due to the limited number of p53 inactivation (deletion/mutation) cases of primary prostate cancers (Supplementary Fig. 1a). These results are consistent with the results from our mouse models in which ERG fusion and Trp53 deletion (Trp53^{-/-}) enabled to induce prostate tumorigenesis in mice (Fig. 1c-e).

Importantly, the co-occurrence of ERG fusion with TP53 shallow deletion (TP53^{WT/-}) and TP53 complete loss (TP53^{-/-}) were highly significantly ($P = 0.0003$ and $P = 0.0075$, respectively) in advanced prostate cancers in the SU2C cohort (Supplementary Fig. 1b). Notably, the co-occurrence of ERG fusion with TP53 deletion/mutation (TP53^{mutation/-}) was most significant ($P = 0.0001$) among all the genotypes examined (Supplementary Fig. 1b), which is consistent with the findings in the GEM models that ERG fusion/Trp53mutant/- mice developed most aggressive tumors in the prostate (Fig. 1c-f). Thus, the data from these new analyses provide further support to our conclusion in this study.

2. In the prostate phenotypic analysis of the genetically modified mice, how was adenocarcinoma scored? The authors should include evidence of invasiveness using epithelial and smooth muscle markers in Figure 1C.

Reply: These are excellent points. The PIN lesions and adenocarcinoma in our genetically modified mice were scored by a GU cancer pathologist (R. Jimenez) using the criteria of mouse PIN/cancer pathology/histology recommended by a consortium (Park et al., *Am J Pathol*, 161: 727-35, 2002) and the studies published by our group (Yang et al., *Cancer Res*, 77: 6524-6537, 2017; Blee et al., *Clin Cancer Res*, 24: 4551-4565, 2018). As suggested by the Reviewer, we performed IHC for CK8 (luminal epithelial cell marker) and smooth muscle actin (SMA) (measuring cell invasiveness) in the benign/prostate cancer tissues of the six groups shown in Fig. 1c. The new data showed that all prostate tissues/prostate tumors are CK8 positive (Supplementary Fig. 2d, Top), indicating that they are all luminal cell types, which is consistent with the observations that all prostate tissues or cancer lesions in these groups of mice are AR positive (Fig. 1c). SMA IHC staining analysis showed that except in prostate tumors of Pb-ERG/Trp53^{R172H/-} mice, SMA expression was intact around the epithelium in other five groups of mice (Supplementary Fig. 2d, Middle). Thus, these data further confirm the invasive phenotype of prostate adenocarcinoma in Pb-ERG/Trp53^{R172H/-} mice.

3. The authors compare the transcriptome from the double ERG and TP53 mutants versus the single genetic alterations. It would be good include the double ERG and TP53 deleted tumours in the validation qRTPCR assays (Figure 2C) to establish if the pathways analysed are divergent between the different TP53 alleles in the presence of ERG alterations.

Reply: As suggested by the Reviewer, we examined the expression of PSGs in the double ERG and TP53 deleted tumors. Our new data showed that upregulation of PSGs was only detected in double ERG/TP53 mutant prostate tumors (Pb-ERG/Trp53^{R172H/-}), but not in double ERG and TP53 deleted tumors (Pb-ERG/Trp53^{-/-}) (Fig. 2g,h). These data provide further support to our conclusion that PSG expression was only specifically elevated in double ERG/p53 mutant prostate tumors.

4. The authors perform a p53 ChIPSeq in VCaP cells, which contain a mutant TP53, and show binding peaks within the CTNNB1 locus. As reports of direct DNA binding of mutant p53 have been varied, it would be good to confirm specificity by performing a ChIP assay on the shp53 cells shown in Fig 3m. The authors mention in the discussion that mutant TP53 has been shown to bind to promoters of nucleotide metabolism genes such as DCK, TK1 and IMPDH1, are these

genes expressed at higher levels in the double mutant prostates and is there evidence of regulation of expression by mutant p53 in VCaP cells in experiments using shp53 cells?

Reply: As suggested by the Reviewer, we performed a ChIP assay on the shp53 cells. We confirmed that p53 knockdown significantly reduced p53 mutant occupancy at the *CTNNB1* gene locus in VCaP cells (Fig. 3m and Supplementary Fig. 5l).

The p53 mutant binding to nucleotide metabolism genes (such as DCK, TK1 and IMPDH1) was reported by an independent group (Do *et al. Genes Dev* **26**, 830-845, 2012) as indicated in the Discussion. As suggested by the Reviewer, we examined expression of DCK, TK1 and IMPDH1 and demonstrated that p53 knockdown did not obviously alter expression of these genes (Supplementary Fig. 4g). Consistent with the gene expression data, ChIP-seq data showed that mutant p53 (R248W) had no occupancy at these gene loci in VCaP cells (Supplementary Table 3). These data suggest that nucleotide metabolism genes such as DCK, TK1 and IMPDH1 are not regulated by mutant p53 in prostate cancer cells, at least in VCaP cells.

5. Beta catenin has been implicated in WNT signalling, therefore some discussion and analysis of how this pathway fits with the model proposed should be done. What are the expression levels of WNT target genes such as AXIN2, LEF1 and MYC in VCaP and DU145 cells knocked out or down for ERG or mutant TP53? Immunohistochemistry for beta catenin should be done on prostates from double mutant animals to define localization within the prostate and within cells. Do patient tumours with mutant TP53 or dysregulated WNT signalling and ERG alterations express higher levels of pyrimide metabolism genes?

Reply: We thank the Reviewer for the great points. As suggested by the Reviewer, we examined expression of WNT target genes such as AXIN2, LEF1 and MYC. We demonstrated that p53 knockdown in both VCaP and DU145 cells reduced the expression of these genes (Supplementary Fig. 6g,h). Notably, expression of these genes was also decreased after knockdown of ERG in ERG fusion-positive VCaP cells (Supplementary Fig. 6g). These data provide further support that the WNT/ β -Catenin signaling pathway is regulated by the p53 pathway.

We also performed IHC analysis of β -Catenin protein expression in prostate normal tissues/cancers in the six groups of GEM mice. Consistent with the mRNA expression data (Fig. 4c), β -Catenin protein was highly expressed in prostate tumors from Pb-ERG/Trp53 RH/- mice, but not in other five groups of mice including Pb-ERG/Trp53 -/- (Supplementary Fig. 2d, Bottom). As shown in Supplementary Fig. 2d, β -Catenin protein was expressed either predominantly in the nucleus of some cells or both cytoplasm and nucleus of most cells in prostate tumors of Pb-ERG/Trp53 RH/- mice,

We also found that high level of CTNNB1 positively correlated with expression of UMPS, RRM1 and RRM2 in SU2C prostate cancer patients (Supplementary Fig. 7l).

6. ERG has been shown to regulate WNT/LEF1 signalling at various levels in prostate cancer, with LEF1 identified as a direct target (Wu et al, Cancer Res 2013). This could mean that the

involvement of ERG in the regulation of pyrimide metabolism genes could be at different levels and this point should be added to the discussion.

Reply: We thank the Reviewer for the great points. We have added this point to the discussion in the revised manuscript on page 26.

7. The authors show that inhibiting the expression of genes involved in pyrimide metabolism can affect growth in VCaP cells in vitro and in vivo. Can nucleotides rescue the growth defect of VCaP cells with shERG and shp53?

Reply: We knocked down endogenous ERG and p53 mutant in VCaP cells and treated cells with or without nucleotides. We found that dATP/dGTP could not effectively rescue the inhibition of growth of VCaP cells induced by knockdown of ERG and p53. In contrast, dTTP/dCTP could significantly rescue the inhibition of VCaP cell growth (Fig. 5h).

8. What other alterations are present in LuCaP 23.1 PDX that might impact the sensitivity in the O'PROTAC compound? As organoids are grown in factors involved in WNT signalling, the drug might not be specific to the TP53 and ERG alterations. This result would be more convincing if the treatment was also done on other organoid lines that do not carry TP53 and ERG alterations.

Reply: This is a good point. It is known that ERG fusion and SPOP mutation are mutually exclusive (TCGA, Cell 163: 1011-25, 2015). Therefore, we generated organoid (PDXO) culture from SPOP Q165P mutation PDX model we established previously (Yan et al., EMBO Mol Med, 11: e10659, 2019). As expected, LEF1 O'PROTAC treatment induced downregulation/degradation of LEF1/TCF proteins (Supplementary Fig. 10a-c). Notably, LEF1 O'PROTAC treatment inhibited the growth of SPOP Q165P PDXO, but the inhibitory effect was not drastic as in ERG fusion/p53 mutation positive LuCaP23.1 PDXO (Fig. 7g, h and Supplementary Fig. 10a-c). Furthermore, Western blot analysis showed that LEF1 O'PROTAC treatment decreased expression of PSGs in ERG-positive LuCaP23.1 PDXO, but not in SPOP Q165P PDXO (Fig. 7f and Supplementary Fig. 10a). In agreement with these observations, addition of dTTP/dCTP had little or no effect on the growth of SPOP Q165P PDXO treated with LEF1 O'PROTAC (Supplementary Fig. 10b, c). These observations are not surprising since SPOP Q165P PDXO do not express ERG fusion which are required to cooperate with β -Catenin to promote expression of PSGs and therefore these organoids are not heavily dependent on pyrimidine synthesis for growth.

Minor point:

The concentrations of O'PROTAC compound in Figure 7 should be mentioned.

Reply: The concentration of O'PROTAC compound used in Figure 7 was 100 nM for treatment of cells in culture and 10 mg/kg/two days for treatment of mice. This information has been added to the figure legends.

Reviewer #4 (Remarks to the Author); expert in nucleotide metabolism:

Remarks to the Author:
Ding and colleagues report the co-occurrence of TMPRSS2-ERG fusion gene with GOF p53 mutations in prostate cancer and associate this genotype to disease progression and aggressiveness. The authors show that ERG and GOF mtp53 cooperatively bind to the promoter of CTNNB1 gene, which in turn activates the expression of b-catenin. As a result, ERG and b-catenin promote the expression of several key enzymes involved in pyrimidine biosynthesis, such as UMPS, RRM1, RRM2 and TYMS, which support the rapid demand for nucleotides of the cancer cells. Thus, pharmacological inhibition of b-catenin by small molecules as well as depletion/degradation by different means of downstream elements, such as LEFT1/TCF or PSG, importantly decrease tumor mass and volume, suggesting a therapeutic target in the treatment of TMPRSS2-ERG/GOF mtp53 prostate cancer. In this context, the manuscript nicely reports a very detailed mechanistic model which could potentially be exploited in the clinics. To this purpose, Ding et al. have included in vitro and in vivo models, as well as a comprehensive set of experiments to prove interaction to the promoter of the different genes and subsequent effects on protein expression and disease progression. As I'm not a specialist in the ChIP-seq analysis here reported, I won't discuss the experimental design, required controls or interpretation of these experiments. Taking that into account I found that overall, the manuscript is well written and contains novel information of relevance in the cancer field.

Reply: We thank the Reviewer for the positivity and recognizing the novelty and significance of our findings in the field. We also thank the Reviewer for the insightful suggestions that have helped us improve our manuscript significantly.

Major points:

- In Fig. 5g a reduction in dCtd and dUrd are observed upon depletion of ERG and p53, interpreted as a depletion in pyrimidine metabolites. Nevertheless, UMPS, RRM1/2 and TYMS are enzymes involved in de novo nucleotide biosynthesis, whereas dCtd and dUrd are either involved in the nucleotide salvage pathways or produced by catabolism of dCTP and dUTP, by the activities of SAMHD1 and dUTPase, respectively. As the aim of this experiment is to demonstrate that shERG and shp53 negatively affect PSG, I would rather analyse the dNTP levels (at least dCTP and dTTP), because they are the final products of this pathway.

Reply: We thank the Reviewer for the excellent point. We repeated metabolite measurement experiments by collecting a new batch of samples and analyzed the level of dNTPs. Consistent with our previous measurement, knockdown of ERG and p53 mutant (R248W) in VCaP cells decreased the level of UMP and dTDP (Fig. 5g), indicating the finding was reproducible. Importantly, our new data showed that the level of dTTP was also decreased after knockdown of ERG and p53 mutant in VCaP cells (Fig. 5g). After checking the original LC-MS data, we did not see any measurement value for dCTP. This could be due to the unique feature of VCaP cells that they grow extremely slow and it is plausible that the dCTP amount in this cell line was under the detectable level by LC-MS and therefore it could be a limiting factor for the low growth of VCaP cells. However, the exact underlying mechanisms warrant systematic investigation in the future.

- The authors show that inhibition of b-catenin by small molecules downregulates mRNA and protein expression of four PSGs in VCaP cells (Fig. 6c,f). However, as described in the manuscript, VCaP is TMPRSS2-ERG/GOF mtp53 double positive and all results are associated to this specific genotype, but no other cell line with a different genotype are shown in this experiment as a negative control. Thus, taking into account that nucleotide biosynthesis is usually upregulated in cancer cells, it could be very interesting to add other prostate cancer cell lines with a different genotype to corroborate whether b-catenin-dependent activation of PSGs are specific to the TMPRSS2-ERG/GOF mtp53 double positive cancer model or if, in contrast, b-catenin could regulate nucleotide levels in other prostate oncological diseases.

Reply: Great suggestions. We examined expression of PSGs in other cell lines with different genotypes (p53 WT and ERG fusion negative) such as LNCaP and C4-2 cells. We found that inhibition of β -Catenin by small molecule inhibitors failed to inhibit expression of PSGs in these two cell lines while expression of the canonical β -Catenin target genes such as CCND1 and c-MYC was largely downregulated (Supplementary Fig. 9c-j). These data further support the notion that β -Catenin-dependent activation of PSGs are regulated by ERG and therefore specific to the TMPRSS2-ERG/GOF mtp53 double positive cancer cells.

- In this manuscript, the authors demonstrate that the consequences of b-catenin inhibition and LEF1/TCF degradation on the tumoral progression are due to the depletion of PGS and thus of pyrimidine nucleotides (Fig 7). Several approved drugs exist targeting the activity of the enzymes encoded by the PGS, such as hydroxyurea and gemcitabine (against RRM), 5-fluorouracil (against TYMS) and pyrazofurin (against UMPS), so supporting the aforementioned hypothesis, I feel very curious to know the effects of those drugs in the development of the disease and whether they could have therapeutical applications in TMPRSS2-ERG/GOF mtp53 double positive prostate cancer.

Reply: These are excellent points. As suggested by the Reviewer, we examined the effect of those drugs on growth of TMPRSS2-ERG/GOF mtp53 double positive prostate cancer VCaP cells. Similar to the previous reports in other cancer types (Fan et al., *Cancer Lett*, 373: 130-7, 2016; Wonganan et al., *Cancer Biol Ther*, 13: 908-14, 2012; Bepler et al., *J Clin Oncol*, 24: 4731-7, 2006), low doses of gemcitabine were ineffective in inhibition of VCaP cell growth. However, we found that growth of VCaP cells was effectively inhibited by high doses of gemcitabine (Supplementary Fig. 8f). Similar results were observed with high doses of hydroxyurea, 5-fluorouracil and pyrazofurin in VCaP cells (Supplementary Fig. 8g-i). These data suggest that these drugs could have therapeutical applications in TMPRSS2-ERG/GOF mtp53 double positive prostate cancer. We have added the new data and discussion in the revised manuscript.

- Last, regarding Fig. 7, though supplementation of dATP/dGTP and dTTP/dCTP revert the phenotype, I have some concerns. Deoxynucleoside triphosphates (dNTPs) are anionic molecules, which prevent passive diffusion over cellular membranes. For this reason, although many chemotherapies exploit the use of nucleotide analogues, they are administered as inactive nucleoside prodrugs, and reversion of dNTP-derived phenotypes are usually achieved through supplementation of nucleosides (in this case deoxyadenosine, deoxyguanosine, thymidine and deoxycytidine). Taking that into account, I didn't find the description of the nucleotide

supplementation (which concentrations were used and the relative ratio of each nucleotide), and believe that the reversion of the phenotype is probably a result of the dNTP degradation to its nucleoside form prior entering the organoids.

Reply: We agree with the Reviewer that dNTPs are anionic molecules and cannot passively diffuse through cellular membranes. Indeed, for our experiments shown in Fig. 7f-h, dNTPs were mixed with control or LEF1 oligonucleotide-PROTACs first and then transfected into LuCaP 23.1 PDXO using lipofectamine. The final concentration of the nucleotides was 10 μ M.

Minor points:

- Line 150, "Upms" should be corrected to "Umps"

Reply: We have corrected the error.

- Line 163, in "with 615 gene", correct to "genes"

Reply: We have corrected the error.

- In Figure 3m, I guess that the third column in the western blot should have a "+" in the shCon row instead of a "-".

Reply: We have corrected the error.

REVIEWERS' COMMENTS

Reviewer #1 (Remarks to the Author):

The authors have address all previous concerns, I have no further comments.

Reviewer #2 (Remarks to the Author):

Acceptable manuscript.

Reviewer #3 (Remarks to the Author):

The authors have addressed my comments in the revised manuscript, which is improved overall compared to the original version.

Minor point:

There is a discrepancy in the P values between the rebuttal comments and the Supplementary Fig1b that needs to be corrected.

Reviewer #4 (Remarks to the Author):

My comments have been nicely addressed in the revised manuscript and I do not have any further comments for this publication.

RESPONSE TO REVIEWERS' COMMENTS

We thank the Reviewers for the time they spent in evaluating our work and for their insightful comments, which we have considered thoroughly in generating the revised manuscript.

REVIEWER COMMENTS

Reviewer #1 (Remarks to the Author):

The authors have addressed all previous concerns, I have no further comments.

Reply: We thank the Reviewer for the positive comments on our current work.

Reviewer #2 (Remarks to the Author):

Acceptable manuscript.

Reply: We thank the Reviewer for the positive comments on our current work.

Reviewer #3 (Remarks to the Author):

The authors have addressed my comments in the revised manuscript, which is improved overall compared to the original version.

Reply: We thank the Reviewer for the positive comments on our current work.

Minor point:

There is a discrepancy in the P values between the rebuttal comments and the Supplementary Fig1b that needs to be corrected.

Reply: We apologize for the carelessness in the last rebuttal comments. We confirm that the co-occurrence of ERG fusion with TP53 shallow deletion (TP53WT/-) was highly significantly ($P = 0.0001$) in advanced prostate cancers in the SU2C cohort (Supplementary Fig. 1b). Therefore, this P value is now consistent with what is shown in the Supplementary Fig 1b.

Reviewer #4 (Remarks to the Author):

My comments have been nicely addressed in the revised manuscript and I do not have any further comments for this publication.

Reply: We thank the Reviewer for the positive comments on our current work.